# INFORMATION BOTTLENECK ANALYSIS OF DEEP NEURAL NETWORKS VIA LOSSY COMPRESSION

**Ivan Butakov**[*,1,2,3], **Alexander Tolmachev**[1,2], **Sofia Malanchuk**[1,2], **Anna Neopryatnaya**[1,2], **Alexey Frolov**[1], **Kirill Andreev**[1]
[1]Skolkovo Institute of Science and Technology; [2]Moscow Institute of Physics and Technology; [3]Sirius University of Science and Technology;
{butakov.id,tolmachev.ad,malanchuk.sv,neopryatnaya.am}@phystech.edu,
{al.frolov,k.andreev}@skoltech.ru

## ABSTRACT

The Information Bottleneck (IB) principle offers an information-theoretic framework for analyzing the training process of deep neural networks (DNNs). Its essence lies in tracking the dynamics of two mutual information (MI) values: between the hidden layer output and the DNN input/target. According to the hypothesis put forth by Shwartz-Ziv & Tishby (2017), the training process consists of two distinct phases: fitting and compression. The latter phase is believed to account for the good generalization performance exhibited by DNNs. Due to the challenging nature of estimating MI between high-dimensional random vectors, this hypothesis was only partially verified for NNs of tiny sizes or specific types, such as quantized NNs. In this paper, we introduce a framework for conducting IB analysis of general NNs. Our approach leverages the stochastic NN method proposed by Goldfeld et al. (2019) and incorporates a compression step to overcome the obstacles associated with high dimensionality. In other words, we estimate the MI between the compressed representations of high-dimensional random vectors. The proposed method is supported by both theoretical and practical justifications. Notably, we demonstrate the accuracy of our estimator through synthetic experiments featuring predefined MI values and comparison with MINE (Belghazi et al., 2018). Finally, we perform IB analysis on a close-to-real-scale convolutional DNN, which reveals new features of the MI dynamics.

## 1 INTRODUCTION

The information-theoretic analysis of deep neural networks (DNNs) is a developing branch of the deep learning theory, which may provide a robust and interpretable way to measure the performance of deep models during training and inference. This type of analysis might complement current non-transparent meta-optimization algorithms for architecture search, like ENAS (Pham et al., 2018), DARTS (Liu et al., 2019; Wu et al., 2019; He et al., 2020), evolutionary algorithms (Fan et al., 2020), and others. This method may also provide new approaches to explainable AI via estimation of information flows in NNs (Tishby & Zaslavsky, 2015; Xu & Raginsky, 2017; Goldfeld et al., 2019; Steinke & Zakynthinou, 2020; Amjad et al., 2022) or via independence testing (Berrett & Samworth, 2017; Sen et al., 2017), as opposed to existing methods of local analysis of a model (Ribeiro et al., 2016; Springenberg et al., 2015; Rs et al., 2020; Ivanovs et al., 2021) or methods based on complex manipulations with data (Lipovetsky & Conklin, 2001; Štrumbelj & Kononenko, 2013; Datta et al., 2016). Information-theoretic quantities can also be considered as regularization terms or training objectives (Tishby & Zaslavsky, 2015; Chen et al., 2016; Belghazi et al., 2018).

The information-theoretic analysis of DNNs relies on the *Information Bottleneck* (IB) principle proposed in Tishby et al. (1999). This concept was later developed in Tishby & Zaslavsky (2015) and applied to DNNs in Shwartz-Ziv & Tishby (2017). The major idea of the IB approach is to track the dynamics of two *mutual information* (MI) values: $I(X; L)$ between the hidden layer output ($L$) and the DNN input ($X$) and $I(Y; L)$ between the hidden layer output and the target of the model ($Y$). As

---

[*]Correspondence to butakov.id@phystech.edu

a result of the IB analysis, the authors of the latter article put forth the so-called *fitting-compression* hypothesis, which states that the training process consists of two phases: a feature-extraction "fitting" phase (both MI values grow) and a representation compression phase ($I(Y; L)$ grows while $I(X; L)$ decreases). The authors conjectured the compression phase to account for the good generalization performance exhibited by DNNs. However, it is still debated whether empirical confirmations of the compression phase are related to improper mutual information estimators, activation function choice, or other implementation details. For a more complete overview of current IB-related problems, we refer the reader to Geiger (2022).

In the original work by Shwartz-Ziv & Tishby (2017), a quantization (or binning) approach was proposed to estimate MI. However, this approach encountered two primary challenges. Firstly, the MI estimate was highly sensitive to the bin size selection. Secondly, for a fixed training epoch, when the training weights are held constant, $L$ becomes a deterministic function of $X$, resulting in the MI being independent of the DNN parameters (and infinite for practically all regimes of interest if we speak about continuous case and reasonable activation functions, see e.g., Amjad & Geiger (2018)). The subsequent papers addressed the aforementioned problems. To tackle the infinite MI problem it was proposed to consider (a) stochastic NNs (Goldfeld et al., 2019; Tang Nguyen & Choi, 2019; Adilova et al., 2023), (b) quantized NNs (Lorenzen et al., 2022) or (c) a mixture of them (Cerrato et al., 2023). Simple and inconsistent binning entropy estimators have been replaced with estimators more appropriate for continuous random variables (Gabrié et al., 2018; Goldfeld et al., 2019; Goldfeld et al., 2020; Adilova et al., 2023).

However, the high-dimensional problem still holds, as the sample complexity (the least number of samples required for an estimation within a fixed additive gap) of any entropy estimator is proven to depend on the dimension exponentially (Goldfeld et al., 2020; McAllester & Stratos, 2020). Due to the challenging nature of estimating MI between high-dimensional random vectors, the fitting-compression hypothesis has only been verified for tiny NNs or special classes of models with tractable information-theoretic quantities (e.g., Gabrié et al. (2018); Lorenzen et al. (2022)). Some existing works on IB-analysis of large networks also exhibit signs of the curse of dimensionality (Goldfeld et al., 2019; Adilova et al., 2023). We mention papers that suggest using lower bounds or other surrogate objectives (Belghazi et al., 2018; Elad et al., 2019; Poole et al., 2019; Darlow & Storkey, 2020; Jónsson et al., 2020; Kirsch et al., 2021; McAllester & Stratos, 2020), advanced binning (Noshad et al., 2019) or even other definitions of entropy (Wickstrøm et al., 2019; Yu et al., 2021) in order to perform IB-analysis of large networks. It may be assumed that these methods can partially overcome the curse of dimensionality via utilizing the internal data structure implicitly, or simply from the fact that non-conventional information theory might be less prone to the curse of dimensionality.

In contrast to the approaches mentioned above, we propose a solution to the curse of dimensionality problem by explicitly compressing the data. Since most datasets exhibit internal structure (according to the *manifold hypothesis* (Fefferman et al., 2013)), it is usually sufficient to estimate information-theoretic quantities using compressed or latent representations of the data. This enables the application of conventional and well-established information-theoretic approaches to real-world machine learning problems. In the recent work of Butakov et al. (2021), the compression was used to obtain the upper bound of the random vector entropy. However, it is necessary to precisely estimate or at least bound from both sides the entropy alternation under compression in order to derive the MI estimate. In the work of Greenewald et al. (2023), two-sided bounds are obtained, but only in the special case of linear compression and smoothed distributions. Our work heavily extends these ideas by providing new theoretical statements and experimental results for MI estimation via compression-based entropy estimation. We stress out the limitations of the previous approaches more thoroughly in the Appendix.

Our contribution is as follows. We introduce a comprehensive framework for conducting IB analysis of general NNs. Our approach leverages the stochastic NN method proposed in Goldfeld et al. (2019) and incorporates a compression step to overcome the obstacles associated with high dimensionality. In other words, we estimate the MI between the compressed representations of high-dimensional random vectors. We provide a theoretical justification of MI estimation under lossless and lossy compression. The accuracy of our estimator is demonstrated through synthetic experiments featuring predefined MI values and comparison with MINE (Belghazi et al., 2018). Finally, the experiment with convolutional DNN classifier of the MNIST handwritten digits dataset (LeCun et al., 2010) is performed. The experiment shows that there may be several compression/fitting phases during the training process. It may be concluded that phases revealed by information plane plots are connected to different regimes of learning (i.e. accelerated, stationary, or decelerated drop of loss function).

It is important to note that stochastic NNs serve as proxies for analyzing real NNs. This is because injecting small amounts of noise have negligible effects on outputs of layers, and the introduced randomness allows for reasonable estimation of information-theoretic quantities that depend on NN parameters. We also mention that injecting noise during training is proven to enhance performance and generalization capabilities (Hinton et al., 2012; Srivastava et al., 2014).

The paper is organized as follows. In Section 2, we provide the necessary background and introduce the key concepts used throughout the paper. Section 3 describes our proposed approach for estimating mutual information under compression, along with theoretical justifications and bounds. In Section 4, we develop a general framework for testing mutual information estimators on synthetic datasets. This framework is utilized in Section 5 to evaluate MINE and four selected mutual information estimators, complemented by the proposed compression step. The best-performing method is then applied in Section 6 to perform information plane analysis on a convolutional NN classifier trained on the MNIST dataset. Finally, the results are discussed in Section 7. We provide all the proofs in the Appendix, as well as discussion of state-of-the-art methods other than MINE.

## 2 PRELIMINARIES

Consider random vectors, denoted as $X : \Omega \to \mathbb{R}^n$ and $Y : \Omega \to \mathbb{R}^m$, where $\Omega$ represents the sample space. Let us assume that these random vectors have absolutely continuous probability density functions (PDF) denoted as $\rho(x)$, $\rho(y)$, and $\rho(x, y)$, respectively, where the latter refers to the joint PDF. The differential entropy of $X$ is defined as follows

$$h(X) = -\mathbb{E} \log \rho(x) = -\int_{\text{supp} X} \rho(x) \log \rho(x) \, dx,$$

where $\text{supp} X \subseteq \mathbb{R}^n$ represents the *support* of $X$, and $\log(\cdot)$ denotes the natural logarithm. Similarly, we define the joint differential entropy as $h(X, Y) = -\mathbb{E} \log \rho(x, y)$ and conditional differential entropy as $h(X \mid Y) = -\mathbb{E} \log \rho(X|Y) = -\mathbb{E}_Y \left( \mathbb{E}_{X|Y=y} \log \rho(X \mid Y = y) \right)$. Finally, the mutual information (MI) is given by $I(X; Y) = h(X) - h(X \mid Y)$, and the following equivalences hold

$$I(X; Y) = h(X) - h(X \mid Y) = h(Y) - h(Y \mid X), \tag{1}$$

$$I(X; Y) = h(X) + h(Y) - h(X, Y). \tag{2}$$

Note that $\text{supp} X$ or $\text{supp} Y$ may have measure zero and be uncountable, indicating a singular distribution. In such cases, if the supports are manifolds, PDFs can be treated as induced probability densities, and $dx$ and $dy$ can be seen as area elements of the corresponding manifolds. Hence, all the previous definitions remain valid.

In the following discussion, we make use of an important property of MI, which is its invariance under non-singular mappings between smooth manifolds. In the next statement we show that the MI can be measured between compressed representations of random vectors.

**Statement 1.** *Let $\xi : \Omega \to \mathbb{R}^{n'}$ be an absolutely continuous random vector, and let $f : \mathbb{R}^{n'} \to \mathbb{R}^n$ be an injective piecewise-smooth mapping with Jacobian $J$, satisfying $n \geq n'$ and $\det \left( J^T J \right) \neq 0$ almost everywhere. Let either $\eta$ be a discrete random variable, or $(\xi, \eta)$ be an absolutely continuous random vector. Then*

$$I(\xi; \eta) = I \left( f(\xi); \eta \right) \tag{3}$$

**Remark 1.** *In what follows by $\xi : \Omega \to \mathbb{R}^{n'}$ we denote the compressed representation of $X$, $n' \leq n$.*

Recall that we utilize the stochastic neural network (NN) approach to address the problem of infinite mutual information $I(X; f(X))$ for a deterministic mapping $f$. As demonstrated in Goldfeld et al. (2019), introducing stochasticity enables proper MI estimation between layers of the network. The stochastic modification of a network serves as a proxy to determine the information-theoretic properties of the original model.

A conventional feedforward NN can be defined as an acyclic computational graph that can be topologically sorted:

$$L_0 \triangleq X, \quad L_1 := f_1(L_0), \quad L_2 := f_2(L_0, L_1), \quad \dots, \quad \hat{Y} \triangleq L_n := f_n(L_0, \dots, L_{n-1}),$$

where $L_0, \ldots, L_n$ denote the outputs of the network's layers. The stochastic modification is defined similarly, but using the Markov chain stochastic model:

**Definition 1.** *The sequence of random vectors $L_0, \ldots, L_n$ is said to form a* stochastic neural network *with input $X$ and output $\hat{Y}$, if $L_0 \triangleq X$, $\hat{Y} \triangleq L_n$, and*

$$L_0 \longrightarrow (L_0, L_1) \longrightarrow \ldots \longrightarrow (L_0, \ldots, L_n)$$

*is a Markov chain; $L_k$ represents outputs of the $k$-th layer of the network.*

Our primary objective is to track $I(L_i; L_j)$ during the training process. In the subsequent sections, we assume the manifold hypothesis to hold for $X$. In such case, under certain additional circumstances (continuity of $f_k$, small magnitude of injected stochasticity) this hypothesis can also be assumed for $L_k$, thereby justifying the proposed method.

## 3 MUTUAL INFORMATION ESTIMATION VIA COMPRESSION

In this section, we explore the application of lossless and lossy compression to estimation of MI between high-dimensional random vectors. We mention the limitations of conventional MI estimators, propose and theoretically justify a complementary lossy compression step to address the curse of dimensionality, and derive theoretical bounds on the MI estimate under lossy compression.

### 3.1 MUTUAL INFORMATION ESTIMATION

Let $\{(x_k, y_k)\}_{k=1}^N$ be a sequence of i.i.d. samples from the joint distribution of random vectors $X$ and $Y$. Our goal is to estimate the mutual information between $X$ and $Y$, denoted as $I(X; Y)$, based on these samples. The most straightforward way to achieve this is to estimate all the components in (1) or (2) via entropy estimators. More advanced methods of MI estimation, like MINE (Belghazi et al., 2018), are also applicable. However, according to Theorem 1 in Goldfeld et al. (2020) and Theorem 4.1 in McAllester & Stratos (2020), sample complexity of entropy (and MI) estimation is exponential in dimension. We show that this obstacle can be overcome if data possesses low-dimensional internal structure.

In our work, we make the assumption of the manifold hypothesis (Fefferman et al., 2013), which posits that data lie along or close to some manifold in multidimensional space. This hypothesis is believed to hold for a wide range of structured data, and there are datasets known to satisfy this assumption precisely (e.g., photogrammetry datasets, as all images are parametrized by camera position and orientation). In our study, we adopt a simplified definition of the manifold hypothesis:

**Definition 2.** *A random vector $X \colon \Omega \to \mathbb{R}^n$ strictly satisfies the manifold hypothesis iff there exist $\xi \colon \Omega \to \mathbb{R}^{n'}$ and $f \colon \mathbb{R}^{n'} \to \mathbb{R}^n$ satisfying the conditions of Statement 1, such that $X = f(\xi)$. A random vector $X' \colon \Omega \to \mathbb{R}^n$ loosely satisfies the manifold hypothesis iff $X' = X + Z$, where $X$ strictly satisfies the manifold hypothesis, and $Z$ is insignificant in terms of some metric.*

To overcome the curse of dimensionality, we propose learning the manifold with autoencoders (Kramer, 1991; Hinton & Salakhutdinov, 2006) and applying conventional estimators to the compressed representations. To address the issue of measure-zero support, we consider the probability measure induced on the manifold.

Let us consider an absolutely continuous $X$, compressible via autoencoder $A = D \circ E$.

**Corollary 1.** *Let $E^{-1} \colon \mathbb{R}^{n'} \supseteq E(\operatorname{supp} X) \to \mathbb{R}^n$ and $E(X) \colon \Omega \to \mathbb{R}^{n'}$ exist, let $(E^{-1} \circ E)(X) \equiv X$, let $Y$, $E(X)$ and $E^{-1}$ satisfy conditions of the Statement 1. Then*

$$I(X; Y) = I(E(X); Y),$$

In case of absolutely continuous $(X, Y)$, the mutual information estimate can be defined as follows:

$$\hat{I}(X; Y) \triangleq \hat{h}(E(X)) + \hat{h}(Y) - \hat{h}(E(X), Y) \tag{4}$$

In case of absolutely continuous $X$ and discrete $Y$, it is impractical to use (2), as the (induced) joined probability distribution is neither absolutely continuous nor discrete. However, (1) is still valid:

$$h(X \mid Y) = \sum_{y \in \operatorname{supp} Y} p_Y(y) \cdot \underbrace{\left[ - \int \rho_X(x \mid Y = y) \log \left( \rho_X(x \mid Y = y) \right) dx \right]}_{h(X \mid Y = y)}$$

Probabilities $p_Y$ can be estimated using empirical frequencies: $\hat{p}_Y(y) = \frac{1}{N} \cdot |\{k \mid y_k = y\}|$. Conditional entropy $h(X \mid Y = y)$ can be estimated using corresponding subsets of $\{x_k\}_{k=1}^N$: $\hat{h}(X \mid Y = y) = \hat{h}(\{x_k \mid y_k = y\})$. The mutual information estimate in this case can be defined as follows:

$$\hat{I}(X;Y) \triangleq \hat{h}(E(X)) - \sum_{y \in \operatorname{supp} Y} \hat{p}_Y(y) \cdot \hat{h}(E(X) \mid Y = y) \tag{5}$$

According to the strong law of large numbers, $\hat{p} \xrightarrow{\text{a.s.}} p$. That is why the convergence of the proposed MI estimation methods solely relies on the convergence of the entropy estimator used in (4) and (5). Note that this method can be obviously generalized to account for compressible $Y$.

## 3.2 Bounds for mutual information estimate

It can be shown that it is not possible to derive non-trivial bounds for $I(E(X);Y)$ in general case if the conditions of Corollary 1 do not hold. Let us consider a simple linear autoencoder that is optimal in terms of mean squared error, which is principal component analysis-based autoencoder. The following statement demonstrates cases where the proposed method of estimating mutual information through lossy compression fails.

**Statement 2.** *For any given $\varkappa \geq 0$ there exist random vectors $X \colon \Omega \to \mathbb{R}^n$, $Y \colon \Omega \to \mathbb{R}^m$, and a non-trivial linear autoencoder $A = D \circ E$ with latent space dimension $n' < n$ that is optimal in terms of minimizing mean squared error $\mathbb{E}\,\|X - A(X)\|^2$, such that $I(X;Y) = \varkappa$ and $I(E(X);Y) = 0$.*

This statement demonstrates that an arbitrary amount of information can be lost through compression of the data. It arises from the fact that "less significant" in terms of metric spaces does not align with "less significant" in terms of information theory. However, with additional assumptions, a more useful theoretical result can be obtained.

Figure 1: Conceptual scheme of Statement 3 in application to lossy compression with autoencoder $A = D \circ E$.

**Statement 3.** *Let $X$, $Y$, and $Z$ be random variables such that $I(X;Y)$ and $I\left((X,Z);Y\right)$ are defined. Let $f$ be a function of two arguments such that $I(f(X,Z);Y)$ is defined. If there exists a function $g$ such that $X = g(f(X,Z))$, then the following chain of inequalities holds:*

$$I(X;Y) \leq I\left(f(X,Z);Y\right) \leq I\left((X,Z);Y\right) \leq I(X;Y) + h(Z) - h(Z \mid X, Y)$$

In this context, $f(X,Z)$ can be interpreted as compressed noisy data, $X$ as denoised data, and $g$ as a perfect denoising decoder. The term $h(Z)$ can be upper-bounded via entropy of Gaussian distribution of the same variance, $h(Z \mid X, Y)$ can be lower-bounded in special cases (e.g., when $Z$ is a sum of independent random vectors, at least one of which is of finite entropy); see Section B of the Appendix for details. We also note the special case where the data lost by compression can be considered as independent random noise.

**Corollary 2.** *Let $X$, $Y$, $Z$, $f$, and $g$ satisfy the conditions of the Statement 3. Let random variables $(X,Y)$ and $Z$ be independent. Then $I(X;Y) = I\left(f(X,Z);Y\right)$.*

We note that (a) the presented bounds cannot be further improved unless additional assumptions are made (e.g., linearity of $f$ in (Greenewald et al., 2023)); (b) additional knowledge about the connection between $X$, $Y$, and $Z$ is required to properly utilize the bounds. Other bounds can also be derived (Sayyareh, 2011; Belghazi et al., 2018; Poole et al., 2019), but they do not take advantage of the compression aspect.

The provided theoretical analysis and additional results from Section B of the Appendix show that the proposed method allows for tracking the true value of MI within the errors of a third-party estimator ran on compressed data and the derived bounds imposed by the compression itself.

## 4 Synthetic dataset generation

In order to test the proposed mutual information estimator, we developed a universal method for synthetic dataset generation with defined information-theoretic properties. This method yields two

random vectors, $X$ and $Y$, with a predefined value of mutual information $I(X;Y)$. The method requires $X$ and $Y$ to be images of normally distributed vectors under known nonsingular smooth mappings. The generation consists of two steps. First, a normal vector $(\xi, \eta) \sim \mathcal{N}(0, M)$ is considered, where $\xi \sim \mathcal{N}(0, I_{n'})$, $\eta \sim \mathcal{N}(0, I_{m'})$, and $n'$, $m'$ are dimensions of $\xi$ and $\eta$, respectively. The covariance matrix $M$ is chosen to satisfy $I(\xi; \eta) = \varkappa$, where $\varkappa$ is an arbitrary non-negative constant.

**Statement 4.** *For every $\varkappa \geq 0$ and every $n', m' \in \mathbb{N}$ exists a matrix $M \in \mathbb{R}^{(n'+m') \times (n'+m')}$ such that $(\xi, \eta) \sim \mathcal{N}(0, M)$, $\xi \sim \mathcal{N}(0, I_{n'})$, $\eta \sim \mathcal{N}(0, I_{m'})$ and $I(\xi; \eta) = \varkappa$.*

After generating the correlated normal random vectors $(\xi, \eta)$ with the desired mutual information, we apply smooth non-singular mappings to obtain $X = f(\xi)$ and $Y = g(\eta)$. According to Statement 1, this step preserves the mutual information, so $I(\xi; \eta) = I(X; Y)$.

## 5 COMPARISON OF THE ENTROPY ESTIMATORS

The MI estimate is acquired according to Subsection 3.1. To estimate the entropy terms in (1) or (2), we leverage conventional entropy estimators, such as kernel density-based (Turlach, 1999; Sayyareh, 2011; Sain, 1994) and Kozachenko-Leonenko estimators (original Kozachenko & Leonenko (1987) and weighted Berrett et al. (2019) versions). To test the accuracy of these approaches, we use datasets sampled from synthetic random vectors with known MI. We generate these datasets in accordance with Section 4.

To examine the impact of the compression step proposed in Subsection 3.1, we utilize a special type of synthetic datasets. Synthetic data lies on a manifold of small dimension. This is achieved by generating a low-dimensional dataset and then embedding it into a high-dimensional space by a smooth mapping (so Statement 1 can be applied). Then, the acquired datasets are compressed via autoencoders. Finally, the obtained results are fed into a mutual information estimator.

Algorithm 1 and Figure 2 describe the proposed mutual information estimation quality measurement. We run several experiments with $f$ and $g$ mapping normal distributions to rasterized images of geometric shapes (e.g., rectangles) or 2D plots of smooth functions (e.g., Gaussian functions).[1] The results are presented in Figures 3 and 4. The blue and green curves correspond to the estimates of MI marked by the corresponding colors in Figure 2. Thus, we see that the compression step does not lead to poor estimation accuracy, especially for the weighted Kozachenko-Leonenko (WKL) estimator, which demonstrates the best performance. Note that we do not plot estimates for uncompressed data, as all the four tested classical estimators completely fail to correctly estimate MI for such high-dimensional data; for more information, we refer to Section E.3 in the Appendix. We also conduct experiments with MINE (without compression), for which we train a critic network of the same complexity, as we use for the autoencoder.

---

**Algorithm 1** Measure mutual information estimation quality on high-dimensional synthetic datasets

---

1: Generate two datasets of samples from normal vectors $\xi$ and $\eta$ with given mutual information as described in section 4 $- \{(x_k, y_k)\}_{k=1}^N$.
2: Choose functions $f$ and $g$ satisfying conditions of Statement 1 (so $I(\xi; \eta) = I(f(\xi); g(\eta))$) and obtain datasets for $f(\xi)$ and $g(\eta) - \{f(x_k)\}_{k=1}^N$, $\{g(y_k)\}_{k=1}^N$.
3: Train autoencoders $A_X = D_X \circ E_X$, $A_Y = D_Y \circ E_Y$ on $\{f(x_k)\}$, $\{g(y_k)\}$ respectively.
4: Obtain datasets for $(E_X \circ f)(\xi)$ and $(E_Y \circ g)(\eta)$. We assume that $E_X$, $E_Y$ satisfy conditions of the Corollary 1, so we expect

$$I(\xi; \eta) = I(f(\xi); g(\eta)) = I\big((E_X \circ f)(\xi); (E_Y \circ g)(\eta)\big)$$

5: Estimate $I\big((E_X \circ f)(\xi); (E_Y \circ g)(\eta)\big)$ and compare the estimated value with the exact one.

---

---

[1]Due to the high complexity of the used $f$ and $g$, we do not define these functions in the main text; instead, we refer to the source code published along with the paper (Butakov et al.).

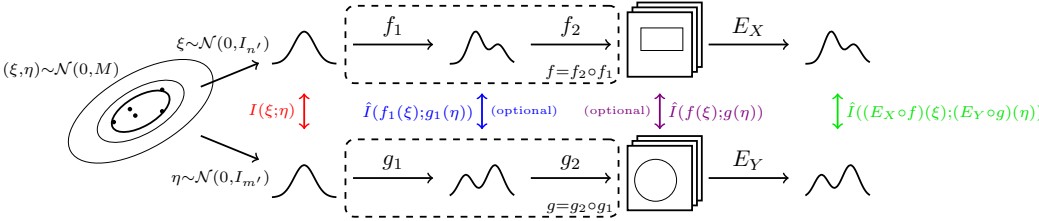

Figure 2: Conceptual scheme of Algorithm 1. In order to observe and quantify the loss of information caused by the compression step, we split $f\colon \mathbb{R}^{n'} \to \mathbb{R}^n$ into two functions: $f_1\colon \mathbb{R}^{n'} \to \mathbb{R}^{n'}$ maps $\xi$ to a structured latent representation of $X$ (e.g., parameters of geometric shapes), and $f_2\colon \mathbb{R}^{n'} \to \mathbb{R}^n$ maps latent representations to corresponding high-dimensional vectors (e.g., rasterized images of geometric shapes). The same goes for $g = g_2 \circ g_1$. Colors correspond to the Figures 3 and 4. For a proper experimental setup, we require $f_1, f_2, g_1, g_2$ to satisfy the conditions of Statement 1.

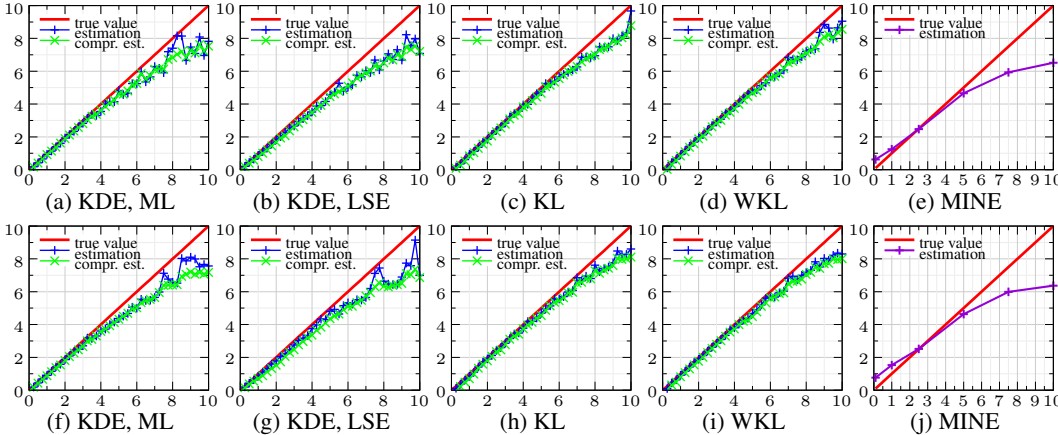

Figure 3: Maximum-likelihood and Least Squares Error KDE, Non-weighted and Weighted Kozachenko-Leonenko, MINE for $16 \times 16$ (first row) and $32 \times 32$ (second row) images of 2D Gaussians ($n' = m' = 2$), $5 \cdot 10^3$ samples. Along $x$ axes is $I(X;Y)$, along $y$ axes is $\hat{I}(X;Y)$.

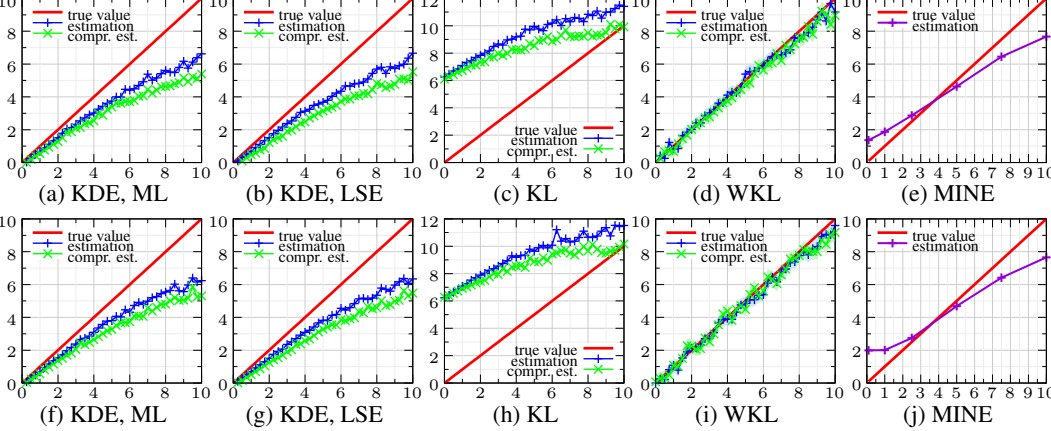

Figure 4: Maximum-likelihood and Least Squares Error KDE, Non-weighted and Weighted Kozachenko-Leonenko, MINE for $16 \times 16$ (first row) and $32 \times 32$ (second row) images of rectangles ($n' = m' = 4$), $5 \cdot 10^3$ samples. Along $x$ axes is $I(X;Y)$, along $y$ axes is $\hat{I}(X;Y)$.

## 6 INFORMATION FLOW IN DEEP NEURAL NETWORKS

This section is dedicated to the information flow estimation in DNNs via the proposed method. We estimate the information flow in a convolutional classifier of the MNIST handwritten digits dataset. This neural network is simple enough to be quickly trained and tested, but at the same time, is complex enough to suffer from the curse of dimensionality. The dataset consists of images of size $28 \times 28 = 784$ pixels. It was shown in Hein & Audibert (2005) that these images have a relatively low latent space dimension, approximately 12–13. If the preservation of only the main features is desired, the latent space can even be narrowed down to 3–10. Although the proposed experimental setup is nowadays considered to be toy and small, it is still problematic for the IB-analysis, as it was shown in Goldfeld et al. (2019).

It can be concluded from the previous section that the weighted Kozachenko-Leonenko estimator is superior to the other methods tested in this paper. That is why it is used in experiments with the DNN classifier described in the current section. The analyzed network is designed to return the output of every layer. To avoid the problem of a deterministic relationship between input and output, we apply Gaussian dropout with a small variance after each layer. This allows for the better generalization during the training (Srivastava et al., 2014) and finite values of MI during the IB-analysis (Adilova et al., 2023). Lossy compression of input images $X$ is performed via a convolutional autoencoder with a latent dimension of $d_X^{\text{latent}}$. Lossy compression of layer outputs $L_i$ is performed via principal component analysis with $d_{L_i}^{\text{latent}}$ as the number of principal components, as it showed to be faster and not significantly worse than general AE approach in this particular case. The general algorithm is described in Algorithm 2.

---

**Algorithm 2** Estimate information flow in the neural network during training

---

1: Compress the input dataset $\{x_k\}_{k=1}^N$ via the input encoder $E_X$: $c_k^X = E_X(x_k)$.
2: **for** epoch : $1, \ldots,$ number of epochs **do**
3:      **for** $L_i$ : layers **do**
4:          Collect outputs of the layer $L_i$: $y_k^{L_i} = L_i(x_k)$. *Each layer must be noisy/stochastic.*
5:          Compress the outputs via the layer encoder $E_{L_i}$: $c_k^{L_i} = E_{L_i}(y_k^{L_i})$.
6:          Estimate $I(E_X(X); E_{L_i}(L_i))$ and $I(E_{L_i}(L_i); Y(X))$, where $Y$ maps inputs to true targets.
7:      **end for**
8:      Perform one-epoch training step of the network.
9: **end for**

---

We use the architecture of the classification network provided in Table 1. We train our network with a learning rate of $10^{-5}$ using the Nvidia Titan RTX. We use $d_X^{\text{latent}} = d_{L_i}^{\text{latent}} = 4$. For other hyperparameters, we refer to Section F of the Appendix and to the source code (Butakov et al.).

| | |
|---|---|
| $L_1$: | Conv2d(1, 8, ks=3), LeakyReLU(0.01) |
| $L_2$: | Conv2d(8, 16, ks=3), LeakyReLU(0.01) |
| $L_3$: | Conv2d(16, 32, ks=3), LeakyReLU(0.01) |
| $L_4$: | Dense(32, 32), LeakyReLU(0.01) |
| $L_5$: | Dense(32, 10), LogSoftMax |

Table 1: The architecture of the MNIST convolution-DNN classifier used in this paper.

The acquired information plane plots are provided in Figure 5. As the direction of the plots with respect to the epoch can be deduced implicitly (the lower left corner of the plot corresponds to the first epochs), we color the lines according to the dynamics of the loss function per epoch. We do this to emphasize one of the key observations: the first transition from fitting to the compression phase coincides with an acceleration of the loss function decrease. It is also evident that there is no clear large-scale compression phase. Moreover, it seems that the number of fitting and compression phases can vary from layer to layer.

## 7 DISCUSSION

An information-theoretic approach to explainable artificial intelligence and deep neural network analysis seems promising, as it is interpretable, robust, and relies on the well-developed information theory. However, the direct application of information-theoretic analysis still poses some problems.

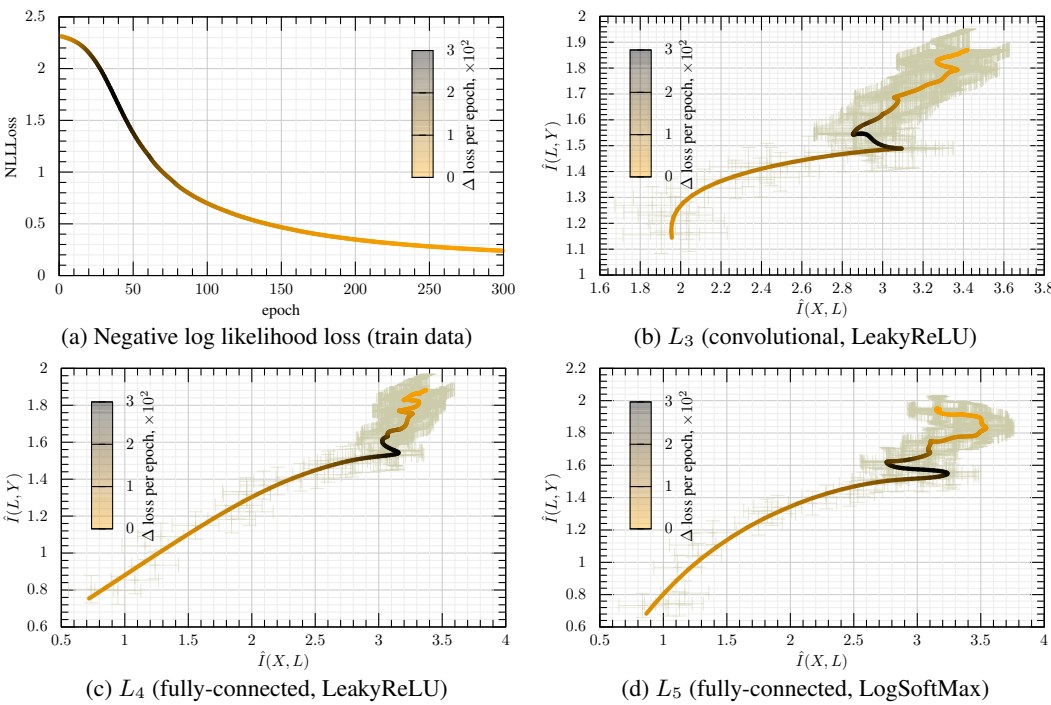

(a) Negative log likelihood loss (train data)

(b) $L_3$ (convolutional, LeakyReLU)

(c) $L_4$ (fully-connected, LeakyReLU)

(d) $L_5$ (fully-connected, LogSoftMax)

Figure 5: Information plane plots for the MNIST classifier. The lower left parts of the plots (b)-(d) correspond to the first epochs. We use 95% asymptotic CIs for the MI estimates acquired from the compressed data. The colormap represents the difference of losses between two consecutive epochs.

We have shown that it is possible to apply information analysis to compressed representations of datasets or models' outputs. To justify our approach, we have acquired several theoretical results regarding mutual information estimation under lossless and lossy compression. These results suggest that this approach is applicable to real datasets. Although it has been shown that an arbitrary amount of information can be lost due to compression, the information required for optimal decompression is still preserved.

We have also developed a framework to test conventional mutual information estimators complemented with the proposed lossy compression step. This framework allows the generation of pairs of high-dimensional datasets with small internal (latent) dimensions and a predefined quantity of mutual information. The conducted numerical experiments have shown that the proposed method performs well, especially when the entropy estimation is done via the weighted Kozachenko-Leonenko estimator. Other methods tend to underestimate or overestimate mutual information.

Finally, an information plane experiment with the MNIST dataset classifier has been carried out. This experiment has shown that the dynamics of information-theoretic quantities during the training of DNNs are indeed non-trivial. However, it is not clear whether the original fitting-compression hypothesis holds, as there is no clear large-scale compression phase after the fitting. We suggest that there may be several compression/fitting phases during the training of real-scale neural networks.

An interesting observation has also been made: the first compression phase coincides with the rapid decrease of the loss function. It may be concluded that the phases revealed by information plane plots are connected to different regimes of learning (i.e., accelerated, stationary, or decelerated drop of the loss function). However, we note that this observation is not the main contribution of our work, and further investigation has to be carried out in order to support this seeming connection with more evidence and theoretical basis.

**Future work.** As further research, we consider using normalizing flows (Rezende & Mohamed, 2015) to improve our approach. Normalizing flows are invertible smooth mappings that provide means of lossless and information-preserving compression. They can be used to transform the joint distribution to a Gaussian, thus facilitating mutual information estimation. Besides, we will apply our method to various large neural networks and perform corresponding information plane analysis.

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

## A COMPLETE PROOFS

**Statement 1.** *Let $\xi\colon \Omega \to \mathbb{R}^{n'}$ be an absolutely continuous random vector, and let $f\colon \mathbb{R}^{n'} \to \mathbb{R}^n$ be an injective piecewise-smooth mapping with Jacobian $J$, satisfying $n \geq n'$ and $\det\left(J^T J\right) \neq 0$ almost everywhere. Let either $\eta$ be a discrete random variable, or $(\xi, \eta)$ be an absolutely continuous random vector. Then*

$$I(\xi; \eta) = I\left(f(\xi); \eta\right) \tag{3}$$

*Proof of Statement 1.* For any function $f$, let us denote $\sqrt{\det\left(J^T(x)J(x)\right)}$ (area transformation coefficient) by $\alpha(x)$ where it exists.

Foremost, let us note that in both cases, $\rho_\xi(x \mid \eta)$ and $\rho_{f(\xi)}(x' \mid \eta) = \rho_\xi(x \mid \eta)/\alpha(x)$ exist. Hereinafter, we integrate over $\operatorname{supp}\xi \cap \{x \mid \alpha(x) \neq 0\}$ instead of $\operatorname{supp}\xi$; as $\alpha \neq 0$ almost everywhere by the assumption, the values of the integrals are not altered.

According to the definition of the differential entropy,

$$h(f(\xi)) = -\int \frac{\rho_\xi(x)}{\alpha(x)} \log\left(\frac{\rho_\xi(x)}{\alpha(x)}\right) \alpha(x)\, dx =$$

$$= -\int \rho_\xi(x) \log\left(\rho_\xi(x)\right) dx + \int \rho_\xi(x) \log\left(\alpha(x)\right) dx =$$

$$= h(\xi) + \mathbb{E}\log\alpha(\xi).$$

$$h(f(\xi) \mid \eta) = \mathbb{E}_\eta\left(-\int \frac{\rho_\xi(x \mid \eta)}{\alpha(x)} \log\left(\frac{\rho_\xi(x \mid \eta)}{\alpha(x)}\right) \alpha(x)\, dx\right) =$$

$$= \mathbb{E}_\eta\left(-\int \rho_\xi(x \mid \eta) \log\left(\rho_\xi(x \mid \eta)\right) dx + \int \rho_\xi(x \mid \eta) \log\left(\alpha(x)\right) dx\right) =$$

$$= h(\xi \mid \eta) + \mathbb{E}\log\alpha(\xi)$$

Finally, by the MI definition,

$$I(f(\xi); \eta) = h(f(\xi)) - h(f(\xi) \mid \eta) = h(\xi) - h(\xi \mid \eta) = I(\xi; \eta).$$

$\square$

**Corollary 1.** *Let $E^{-1}\colon \mathbb{R}^{n'} \supseteq E(\operatorname{supp}X) \to \mathbb{R}^n$ and $E(X)\colon \Omega \to \mathbb{R}^{n'}$ exist, let $(E^{-1} \circ E)(X) \equiv X$, let $Y$, $E(X)$ and $E^{-1}$ satisfy conditions of the Statement 1. Then*

$$I(X; Y) = I(E(X); Y),$$

*Proof of Corollary 1.*

$$I(X; Y) = \underbrace{I\left((E^{-1} \circ E)(X); Y\right) = I\left(E(X); Y\right)}_{\text{from the Statement 1}}$$

$\square$

**Statement 2.** *For any given $\varkappa \geq 0$ there exist random vectors $X\colon \Omega \to \mathbb{R}^n$, $Y\colon \Omega \to \mathbb{R}^m$, and a non-trivial linear autoencoder $A = D \circ E$ with latent space dimension $n' < n$ that is optimal in terms of minimizing mean squared error $\mathbb{E}\left\|X - A(X)\right\|^2$, such that $I(X; Y) = \varkappa$ and $I(E(X); Y) = 0$.*

*Proof of Statement 2.* Let us consider the following three-dimensional Gaussian vector $(X_1, X_2, Y) \triangleq (X, Y)$:

$$X \sim \mathcal{N}\left(0, \begin{bmatrix} 1 & 0 \\ 0 & \sigma \end{bmatrix}\right), \qquad Y \sim \mathcal{N}(0, 1) \qquad (X_1, X_2, Y) \sim \mathcal{N}\left(0, \begin{bmatrix} 1 & 0 & 0 \\ 0 & \sigma & a \\ 0 & a & 1 \end{bmatrix}\right),$$

where $\operatorname{cov}(X_2, Y) = a \triangleq \sqrt{1 - e^{-2\varkappa}}$, $\operatorname{cov}(X_1, Y) = 0$ (so $X_1$ and $Y$ are independent). Let the intrinsic dimension be $n' = 1$, and $\sigma < 1$. According to the principal component analysis, the optimal linear encoder is defined up to a scalar factor by the equality $E(X) = X_1$. However, $I(X; Y) = -\frac{1}{2}\ln\left(1 - a^2\right) = \varkappa$ (see Statement 5), but $I(E(X); Y) = 0$, as $X_1$ and $Y$ are independent. $\square$

**Statement 3.** *Let $X$, $Y$, and $Z$ be random variables such that $I(X;Y)$ and $I((X,Z);Y)$ are defined. Let $f$ be a function of two arguments such that $I(f(X,Z);Y)$ is defined. If there exists a function $g$ such that $X = g(f(X,Z))$, then the following chain of inequalities holds:*

$$I(X;Y) \leq I\left(f(X,Z);Y\right) \leq I((X,Z);Y) \leq I(X;Y) + h(Z) - h(Z \mid X,Y)$$

*Proof of Statement 3.* According to data processing inequality (Cover & Thomas, 2006), $I\left(f(X,Z);Y\right) \leq I(X,Z;Y)$. As $I(X;Y) = I\left(g(f(X,Z));Y\right)$, $I(X;Y) \leq I\left(f(X,Z);Y\right)$.

Note that as DPI is optimal, additional assumptions on $f$, $X$, $Y$ and $Z$ are required to tighten the bounds.

The last inequality is derived via the following equations from Cover & Thomas (2006):

$$I(X,Z;Y) = I(X;Y) + I(Y;Z \mid X)$$
$$I(X,Y;Z) = I(X;Z) + I(Y;Z \mid X)$$

As $I(X;Z) \geq 0$,

$$I(X;Y) + I(X,Y;Z) = I(X;Y) + I(Y;Z \mid X) + I(X;Z) \geq$$
$$\geq I(X;Y) + I(Y;Z \mid X) = I(X,Z;Y)$$

Finally, recall that $I(X,Y;Z) = h(Z) - h(Z \mid X,Y)$. $\qquad\square$

**Corollary 2.** *Let $X$, $Y$, $Z$, $f$, and $g$ satisfy the conditions of the Statement 3. Let random variables $(X,Y)$ and $Z$ be independent. Then $I(X;Y) = I\left(f(X,Z);Y\right)$.*

*Proof of Corollary 2.* Since $(X,Y)$ and $Z$ are independent, $I\left(X,Z;Y\right) = I(X;Y)$, which implies $I(X;Y) = I\left(f(X,Z);Y\right)$ according to the Statement 3. $\qquad\square$

**Statement 4.** *For every $\varkappa \geq 0$ and every $n', m' \in \mathbb{N}$ exists a matrix $M \in \mathbb{R}^{(n'+m') \times (n'+m')}$ such that $(\xi, \eta) \sim \mathcal{N}(0, M)$, $\xi \sim \mathcal{N}(0, I_{n'})$, $\eta \sim \mathcal{N}(0, I_{m'})$ and $I(\xi; \eta) = \varkappa$.*

*Proof of Statement 4.* We divide the proof into the following statements:

**Statement 5.** *Let $(\xi, \eta) \sim \mathcal{N}(0, M)$ be a Gaussian pair of (scalar) random variables with unit variance such that $I(\xi; \eta) = \varkappa$. Then*

$$M = \begin{bmatrix} 1 & a \\ a & 1 \end{bmatrix}, \qquad a = \sqrt{1 - e^{-2\varkappa}} \tag{6}$$

*Proof.* Differential entropy of multivariate normal distribution $\mathcal{N}(\mu, \Sigma)$ is $h = \frac{1}{2} \ln\left(\det\left(2\pi e \cdot \Sigma\right)\right)$. This and (2) leads to the following:

$$\varkappa = I(\xi; \eta) = \frac{1}{2} \ln(2\pi e) + \frac{1}{2} \ln(2\pi e) - \frac{1}{2} \ln\left((2\pi e)^2 \cdot (1 - a^2)\right) = -\frac{1}{2} \ln\left(1 - a^2\right)$$

$$a = \sqrt{1 - e^{-2\varkappa}}$$

$\qquad\square$

**Statement 6.** *Let $\xi$ and $\eta$ be independent random variables. Then $I(\xi; \eta) = 0$, $h(\xi, \eta) = h(\xi) + h(\eta)$.*

*Proof.* We consider only the case of absolutely continuous $\xi$. As $\xi$ and $\eta$ are independent, $\rho_\xi(x \mid \eta = y) = \rho_\xi(x)$. That is why $I(\xi; \eta) = h(\xi) - h(\xi \mid \eta) = h(\xi) - h(\xi) = 0$, according to the definition of MI. The second equality is derived from (2). $\qquad\square$

**Corollary 3.** *Let $\xi_1$, $\xi_2$ and $\eta_1$, $\eta_2$ be random variables, independent in the following tuples: $(\xi_1, \xi_2)$, $(\eta_1, \eta_2)$ and $((\xi_1, \eta_1), (\xi_2, \eta_2))$. Then $I\left((\xi_1, \xi_2); (\eta_1, \eta_2)\right) = I(\xi_1; \eta_1) + I(\xi_2; \eta_2)$*

*Proof.* From (2) and Statement 6 the following chain of equalities is derived:

$$I\left((\xi_1, \xi_2); (\eta_1, \eta_2)\right) = h(\xi_1, \xi_2) + h(\eta_1, \eta_2) - h(\xi_1, \xi_2, \eta_1, \eta_2) =$$
$$= h(\xi_1) + h(\xi_2) + h(\eta_1) + h(\eta_2) - h(\xi_1, \eta_1) - h(\xi_2, \eta_2) =$$
$$= I(\xi_1; \eta_1) + I(\xi_2; \eta_2)$$

$\square$

The Statement 5 and Corollary 3 provide us with a trivial way of generating dependent normal random vectors with a defined mutual information. Firstly, we consider $\Xi \sim \mathcal{N}(0, M')$, where $M'$ is a $(n' + m') \times (n' + m')$ block-diagonal matrix with blocks from (6). The number of blocks is $k = \min\{n', m'\}$ (other diagonal elements are units). The parameter $\varkappa$ for each block equals $I(\xi; \eta)/k$, where $I(\xi; \eta)$ is the desired mutual information of the resulting vectors. The components of $\Xi$ are then rearranged to get $(\xi, \eta) \sim \mathcal{N}(0, M)$, where $\xi \sim \mathcal{N}(0, I_{n'})$ and $\eta \sim \mathcal{N}(0, I_{m'})$. The final structure of $M$ is as follows:

$$M = \left[\begin{array}{cccc|cccc} 1 & & & & a & & & \\ & 1 & & & & a & & \\ & & \ddots & & & & \ddots & \\ \hline a & & & & 1 & & & \\ & a & & & & 1 & & \\ & & \ddots & & & & \ddots & \end{array}\right] \begin{array}{c} \\ \underbrace{\phantom{xxxx}}_{n'} \underbrace{\phantom{xxxx}}_{m'} \end{array} \tag{7}$$

$\square$

## B    ENTROPY BOUNDS

In this section, we provide several theoretical results that complement the bounds proposed in Section 3.2. The following inequalities can be used to bound the entropy terms in Statement 3.

**Statement 7** (Cover & Thomas (2006), Theorem 8.6.5). *Let $X$ be a random vector with covariance matrix $R$. Then $h(X) \leq h(\mathcal{N}(0, R))$.*

**Statement 8.** *Let $X, Z \colon \Omega \to \mathbb{R}^n$ be independent random vectors. Then $h(X + Z) \geq h(Z)$.*

*Proof.* Recall that

$$h(X, X + Z) = h(X + Z) + h(X \mid X + Z) = h(X) + h(X + Z \mid X),$$

from which the following is derived:

$$h(X + Z) = h(X) + h(X + Z \mid X) - h(X \mid X + Z)$$

Note that $h(X + Z \mid X) = \mathbb{E}_X h(x + Z \mid X = x) = h(Z \mid X)$. As $X$ and $Z$ are independent, $h(Z \mid X) = h(Z)$. Thus, we derive the following:

$$h(X + Z) = h(X) + h(Z) - h(X \mid X + Z) = h(Z) + \underbrace{I(X; X + Z)}_{\geq 0} \geq h(Z)$$

$\square$

**Statement 9.** *Let $X \colon \Omega \to \mathbb{R}^{n \times n}$ and $Z \colon \Omega \to \mathbb{R}^n$ be a random matrix and vector, correspondingly. Let $X$ and $Z$ be independent. Then $h(X \cdot Z) \geq h(Z) + \mathbb{E}\left(\ln|\det X|\right)$.*

*Proof.* Note that $h(X \cdot Z \mid X) = \mathbb{E}_X h(x \cdot Z \mid X = x) = h(Z \mid X) + \mathbb{E}\left(\ln|\det X|\right)$. The rest of the proof is the same as for Statement 8:

$$h(X \cdot Z) = h(X) + h(Z) + \mathbb{E}\left(\ln|\det X|\right) - h(X \mid X \cdot Z) =$$
$$= h(Z) + \underbrace{I(X; X \cdot Z)}_{\geq 0} + \mathbb{E}\left(\ln|\det X|\right) \geq h(Z) + \mathbb{E}\left(\ln|\det X|\right).$$

$\square$

**Corollary 4.** *Let $X, Z \colon \Omega \to \mathbb{R}^n$ be independent random vectors. Then $h(X \odot Z) \geq h(Z) + \sum_{i=1}^{n} \mathbb{E}\left(\ln |X_i|\right)$, where $\odot$ is an element-wise product.*

*Proof.* Note that $X \odot Z = \operatorname{diag}(X) \cdot Z$, and $\log |\det \operatorname{diag}(X)| = \sum_{i=1}^{n} \ln |X_i|$. We then apply Statement 9. $\qquad\square$

Note that entropy terms in Statements 8 and 9 can be conditioned. The independence requirement should then be replaced by independence under corresponding conditions.

We also note that Statement 7 can utilize autoencoder reconstruction error (via error covariance matrix), and Statements 8, 9 – magnitude of random vector and injected noise, which is of particular use, as this information is easily accessible in a typical experimental setup.

Practical use cases include using Statement 8 when stochasticity is introduced via additive noise (e.g., Goldfeld et al. (2019)) and Corollary 4 when stochasticity is introduced via multiplicative noise (e.g., Adilova et al. (2023)).

## C  LIMITATIONS OF PREVIOUS WORKS

In this section, we stress the novelty of our contribution to the problem of high-dimensional MI estimation.

In the Introduction, we mention the works of Butakov et al. (2021) and Greenewald et al. (2023). The first article focuses on entropy estimation via lossy compression. The main theoretical result of the paper in question is the following upper bound of random vector entropy:

**Statement 10** (Butakov et al. (2021)). *Let $X$ be a random vector of dimension $n$, let $Z \sim \mathcal{N}(0, \sigma^2 I_{n'})$. Let $A = D \circ E$ be an autoencoder of input dimension $n$ and latent dimension $n'$. Then*

$$h(X) \leq h(E(X) + Z) - n'\left(c + \frac{\log(\sigma^2)}{2}\right) + n\left(c + \frac{\log(\Sigma^2)}{2}\right),$$

*where*

$$c = \frac{\log(2\pi e)}{2}, \quad \Sigma^2 = \frac{1}{n}\mathbb{E}\left[\|X - D(E(X) + Z)\|^2\right]$$

This bound takes advantage of the compression aspect, as it incorporates the reconstruction mean squared error $\Sigma^2$. However, it is important to note several limitations of the proposed bound. Firstly, this Statement is insufficient to acquire any bound of MI, as MI is computed via difference of entropy terms (see (1) and (2)), so a two-sided bound of entropy is required. Secondly, this bound is derived in case of additive Gaussian noise being injected into the latent representation of the vector. It is inapplicable to other cases of stochasticity injection (e.g., noise added to the vector itself) and, moreover, deteriorates when $\sigma \to 0$. That is why we consider this result inapplicable to the task of MI estimation in the current form.

Now, consider the following two-sided bound derived in the work of Greenewald et al. (2023):

**Statement 11** (Greenewald et al. (2023)). *Let $X$ be a random vector of dimension $n$, let $Z \sim \mathcal{N}(0, \sigma^2 I_n)$. Let $E$ be a PCA-projector to a linear manifold of dimension $n'$ with explained variances denoted by $\lambda_i$ in the descending order. Then*[2]

$$\frac{n - n'}{2}\log\left(2\pi e \sigma^2\right) \leq h(X + Z) - h(E(X + Z)) \leq \frac{n - n'}{2}\log\left(2\pi e(\lambda_{n'+1} + \sigma^2)\right)$$

This bound also takes advantage of the compression aspect, as it incorporates the reconstruction mean squared error via $\lambda_i$. We also note that, as the bound is two-sided, corresponding bounds of MI estimate under Gaussian convolution and linear compression can be derived:

**Corollary 5.** *Under the conditions of Statement 11*

$$|I(X + Z; Y) - I(E(X + Z); Y)| \leq \frac{n - n'}{2}\log\left(1 + \frac{\lambda_{n'+1}}{\sigma^2}\right)$$

---

[2]We believe "$+Z$" in "$h(E(X + Z))$" to be missing in the original article; counterexample: $X = const$.

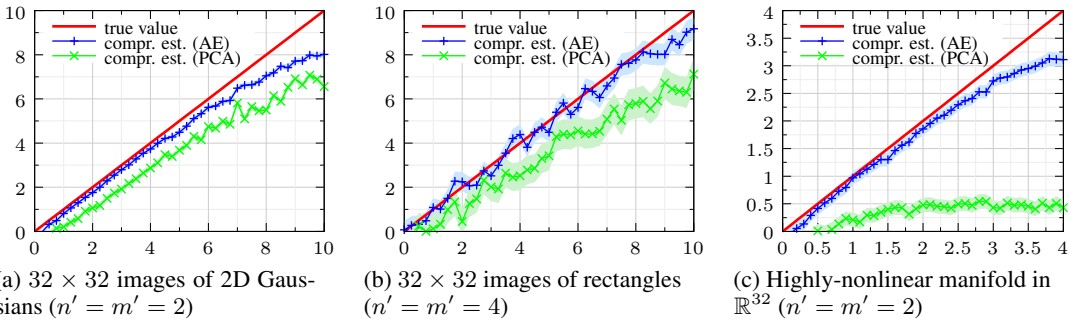

(a) $32 \times 32$ images of 2D Gaussians ($n' = m' = 2$)

(b) $32 \times 32$ images of rectangles ($n' = m' = 4$)

(c) Highly-nonlinear manifold in $\mathbb{R}^{32}$ ($n' = m' = 2$)

Figure 6: Comparison of nonlinear AE and linear PCA performance in task of MI estimation via lossy compression, $5 \cdot 10^3$ samples. Along $x$ axes is $I(X;Y)$, along $y$ axes is $\hat{I}(X;Y)$. WKL entropy estimator is used

*Proof.* From the (2) we acquire

$$I(X + Z;Y) - I(E(X + Z);Y) =$$
$$= [h(X + Z) - h(E(X + Z))] - [h(X + Z, Y) - h(E(X + Z), Y)]$$

Bounds from Statement 11 can be applied to the joint entropy, as $Y$ can be viewed as additional components of the vector $X$, unaffected by smoothing and compression.

To acquire the upper bound of difference of MI terms, we apply the upper bound from Statement 11 to the first difference of entropy terms and the lower bound to the second, and vice versa in case of the lower bound. After simplification, we acquire the desired formula. □

Corollary 5 provides a useful result, as (a) the difference between the true MI and MI under compression is bounded, (b) the bound converges to zero as $\lambda_{n'+1} \to 0$ (which corresponds to the lossless compression). However, as the authors of the original paper mention, this bound deteriorates when $\sigma \to 0$, which coincides with Statement 2.

The linearity of the encoder $E$ is another limitation we have to mention. Although the possibility of extension to nonlinear dimensionality reduction approaches is mentioned in the paper, it is unclear if the derived bounds could be directly transferred to the nonlinear case. To accomplish this, one has to propose a nonlinear generalization of explained variance and provide a more general analysis of entropy alternation via discarding nonlinear components. We also perform tests with synthetic data to show that autoencoder-based approach outperforms PCA-based in case of nonlinear manifolds (see Figure 6). Although the gap is relatively small for the datasets we used in Section 4, it is possible to provide an example of a highly nonlinear manifold, in which case the linear compression is very lossy (see Figure 6c, the synthetic data generator is provided in the source code).

Finally, we note that similar, or even tighter bounds can be derived from the theoretical results of our work.

**Corollary 6.** *Under the conditions of Statement 11 the following inequalities hold:*

$$0 \leq I(X + Z;Y) - I(E(X + Z);Y) \leq \frac{n - n'}{2} \log\left(1 + \frac{\lambda_{n'+1}}{\sigma^2}\right)$$

*Proof.* To avoid notation conflicts, we denote $X$ and $Z$ used in Statement 3 as $X'$ and $Z'$ correspondingly. In order to simplify the following analysis, we consider $E: \mathbb{R}^n \to \mathbb{R}^n$ (ran $E \subseteq \mathbb{R}^{n'}$ is embedded in $\mathbb{R}^n$) and $E \circ E = E$, $E^T = E$ (PCA is used in the form of an orthogonal projector). We then choose $f = E$, $g = \mathrm{Id}$, $X' = E(X + Z)$, $Z' = X + Z - E(X + Z)$ (so $Z' \perp X'$), which yields the following inequalities:

$$I(X';Y) \leq I(X', Z';Y) \leq I(X';Y) + h(Z') - h(Z' \mid X', Y)$$

We then utilize Statements 7 and 8 to bound the entropy terms:

$$h(Z') \leq \frac{n - n'}{2} \log\left(2\pi e(\lambda_{n'+1} + \sigma^2)\right), \qquad h(Z' \mid X', Y) \geq \frac{n - n'}{2} \log\left(2\pi e\sigma^2\right)$$

Thus, the following result is acquired:

$$I(X';Y) \leq I(X',Z';Y) \leq I(X';Y) + \frac{n-n'}{2} \log\left(1 + \frac{\lambda_{n'+1}}{\sigma^2}\right)$$

Now recall that $X' = E(X+Z)$, $(X',Z') \sim X' + Z' = X + Z$. This yields the final result. $\quad\square$

## D    LIMITATIONS OF OTHER ESTIMATORS

In this section, we provide a brief overview of modern entropy and MI estimators that achieve a relative practical success in dealing with the curse of dimensionality. We provide reasoning why we choose MINE (Belghazi et al., 2018) as the only modern MI estimator among the mentioned in the Introduction to compare our results with.

- **MINE** is widely considered as a good benchmark estimator and featured in several recent works (Poole et al., 2019; Jónsson et al., 2020; McAllester & Stratos, 2020; Mroueh et al., 2021). As MINE is a neural estimator, it is theoretically able to grasp latent structure of data, thus performing compression implicitly.
- **Other lower/upper bounds and surrogate objectives.**
    - According to McAllester & Stratos (2020), not many methods in question outperform MINE. In fact, among the other methods mentioned in McAllester & Stratos (2020), only the difference of entropies (DoE) estimator achieves good results during a standard correlated Gaussians test. Unfortunately, DoE requires good parametrized and differentiable (by parameters) estimates of two PDFs, which is difficult to achieve in the case of complex multidimensional distributions.
    - According to an another overview (Poole et al., 2019), the methods in question have various significant trade-offs. Some of them require parts of the original distribution (like $\rho_{Y|X}$) or even some information-theoretic quantities (like $h(X)$) to be tractable, which is not achievable without utilizing special kinds of stochastic NNs. The others heavily rely on fitting a critic function to partially reproduce the original distribution, which leads to a poor bias-variance trade-off (it is illustrated by the results of these estimators in a standard correlated Gaussians test, see Figure 2 in Poole et al. (2019)).
    - Compared to autoencoders, critic networks in methods in question are usually unstable and hard to train, see experiments in Poole et al. (2019); McAllester & Stratos (2020). We also have witnessed this instability while conducting experiments with MINE, see the attached source code.

  We, however, note that all these methods are of great use for building information-theoretic training objectives (as they are differentiable and usually represent upper or lower bounds).

  In addition to the limitations mentioned above, we would like to note that the relative practical success of the modern NN-based MI estimators might be attributed to the data compression being performed implicitly.

  In the work of Poole et al. (2019) it has been shown that other complex parametric NN-based estimators (NJW, JS, InfoNCE, etc.) exhibit poor performance during the estimation of MI between a pair of 20-dimensional incompressible (i.e., not lying along a manifold) synthetic vectors. These vectors, however, are of much simpler structure than the synthetic datasets used in our work (Gaussian vectors and $x \mapsto x^3$ mapping applied to Gaussian vectors in Poole et al. (2019) versus high-dimensional images of geometric shapes and functions in our work). We interpret this phenomenon as a practical manifestation of the universal problem of MI estimation, which also affects the performance of modern NN-based MI estimators in the case of hard-to-compress data.

- **EDGE** (Noshad et al., 2019) is a generalized version of the original binning estimator proposed in Shwartz-Ziv & Tishby (2017): the binning operation is replaced by a more general hashing. We suppose that this method suffers from the same problems revealed in Goldfeld et al. (2019), unless a special hashing function admitting manifold-like or cluster-like structure of complex high-dimensional data is used.
- **Other definitions of entropy**. We are interested in fitting-compression hypothesis (Tishby & Zaslavsky, 2015; Shwartz-Ziv & Tishby, 2017) which is formulated for the classical

mutual information, so other definitions are not appropriate for this particular task. We also note that the classical theory of information is well-developed and provides rigorous theoretical results (e.g., data processing inequality, which we used to prove Statement 3).

- We also mention the approach proposed in Adilova et al. (2023), where $h(L \mid X)$ is computed via a closed-form formula for Gaussian distribution and Monte-Carlo sampling. However, we note the following drawbacks of this method: (a) a closed-form formula is applicable to the entropy estimation only for the first stochastic NN layer, (b) a general-case estimator still has to be utilized to estimate $h(L)$ (in the work of Adilova et al. (2023), the plug-in estimator from Goldfeld et al. (2019) is used; this estimator also suffers from the curse of dimensionality).

# E  CLASSICAL ENTROPY ESTIMATORS

In this section, we provide definitions of conventional entropy estimators used to conduct the experiments, as well as provide proofs that these estimators fail in case of high-dimensional data.

## E.1  KERNEL DENSITY ESTIMATION

The estimation of the probability density function for codes $c_k$ in the latent space plays an important role in the proposed method of mutual information estimation. There are many methods for probability density function estimation (e.g., Weglarczyk (2018); Kozachenko & Leonenko (1987); Berrett et al. (2019)). One of the most popular methods for solving this problem is kernel density estimation (KDE). In this section, we study this method in application to entropy estimation.

Let $\rho_{X,k}(x)$ be a density estimate at a point $x$, which is obtained from the sampling $\{x_k\}_{k=1}^N$ without the $k$-th element by KDE with the kernel $K$. We get the following expression for the density:

$$\hat{\rho}_{b,-k}(x) = \frac{1}{b^n (N-1)} \sum_{\substack{l=0 \\ l \neq k}}^{N} K\left(\frac{x - x_l}{b}\right) \tag{8}$$

Here, $K(x) = \frac{1}{(2\pi)^{n/2}} \exp\left(-\frac{\|x\|^2}{2}\right)$ is a standard Gaussian kernel.[3]

The entropy estimate is obtained via Leave-One-Out method. Densities at each sample $x_k$ are calculated according to the formula 8.

$$\hat{H}(X) = \frac{1}{N} \sum_{k=1}^{N} \log \hat{\rho}_{b,-k}(x_k) \tag{9}$$

### E.1.1  MAXIMUM-LIKELIHOOD

The optimal bandwidth can be selected in accordance with the minimization of the Kullback-Leibler divergence between the estimated distributions and the empirical one ($\hat{\rho}_{\text{emp}}(x) = \frac{1}{N} \sum_{k=1}^{N} \delta(x - x_k)$). This is equivalent to selecting the bandwidth as a maximum likelihood estimate:

$$\hat{b} = \arg\max_b \hat{H}(X) = \arg\max_b \sum_{k=1}^{N} \log \hat{\rho}_{b,-k}(x_k) \tag{10}$$

The experiments have shown that this method tends to underestimate mutual information, and the difference increases with an increasing true value of mutual information.

Asymptotic: *the entropy estimation and bandwidth selection take $\mathcal{O}(n \log n)$, compression takes $\mathcal{O}(n)$, resulting in a total time complexity of $\mathcal{O}(n \log n)$*

---

[3]Hereinafter, it is possible to use any kernel with infinite support as $K$, but the Gaussian one is preferable because of its light tails and infinite differentiability.

E.1.2 LEAST SQUARES ERROR

Now let us consider the Least Square Cross Validation method (see Turlach (1999); Sain (1994)). In this method, bandwidth selection is based on the minimization of the mean squared error between the exact density and the corresponding kernel density estimate. We minimize the following expression:

$$ISE(b) = \int\limits_{\mathbb{R}^n} \left( \hat{\rho}_b(x) - \rho(x) \right)^2 dx$$

Here, $\rho$ is the true probability density function, and $\hat{\rho}_b(x)$ is the estimate with the bandwidth $b$:

$$\hat{\rho}_b(x) = \frac{1}{b^n N} \sum_{k=1}^{N} K\left( \frac{x - x_k}{b} \right)$$

Since the true distribution is unknown, we substitute $\rho$ with $\hat{\rho}_{\text{emp}}$. This leads to the following objective function to be minimized:

$$\frac{1}{N^2} \sum_{i=1}^{N} \sum_{j=1}^{N} J_b(x_i - x_j) - \frac{2}{N} \sum_{i=1}^{N} \hat{\rho}_{b,-i}(x_i),$$

where

$$J_b(\xi) = \int\limits_{\mathbb{R}^n} \frac{1}{b^{2n}} K\left( \frac{x}{b} \right) K\left( \frac{x - \xi}{b} \right) dx,$$

which can be computed via the Fourier transform.

Asymptotic: *The entropy estimation takes $\mathcal{O}\left( n \log n \right)$, compression takes $\mathcal{O}\left( n \right)$, same as KDE ML. However, the optimal bandwidth selection takes $\mathcal{O}\left( n^2 \right)$ due to the quadratic complexity of the minimized objective. Therefore, this algorithm has a total time complexity of $\mathcal{O}\left( n^2 \right)$, making KDE LSE asymptotically the slowest algorithm implemented within this research.*

E.2 KOZACHENKO-LEONENKO

There is another method of entropy estimation, which was proposed by Kozachenko and Leonenko in Kozachenko & Leonenko (1987). The main feature of this method is that it utilizes $k$-nearest neighbor density estimation instead of KDE.

E.2.1 NON-WEIGHTED KOZACHENKO-LEONENKO

Let $\{x_k\}_{k=1}^{N} \subseteq \mathbb{R}^n$ be the sampling of random vector $X$. Let us denote $\hat{r}(x) = \min\limits_{1 \leq k \leq N} r(x, x_k)$ the distance to the nearest neighbour using the metric $r$ (by default, $r$ is Euclidean metric).

According to Kozachenko & Leonenko (1987), the density estimation at $x$ is given by:

$$\hat{\rho}(x) = \frac{1}{\gamma \cdot \hat{r}(x)^n \cdot c_1(n) \cdot (N-1)},$$

where $c_1(n) = \pi^{n/2} / \Gamma(n/2 + 1)$ is a unit $n$-dimensional ball volume and $\gamma$ is a constant which makes the entropy estimate unbiased ($\ln \gamma = c_2 \approx 0.5772$ is the Euler constant).

Asymptotic: *the entropy estimation takes $\mathcal{O}\left( n \log n \right)$, compression takes $\mathcal{O}\left( n \right)$, resulting in a total time complexity of $\mathcal{O}\left( n \log n \right)$.*

E.2.2 WEIGHTED KOZACHENKO-LEONENKO

The main drawback of the conventional Kozachenko-Leonenko estimator is the bias that occurs in dimensions higher than 3. This issue can be addressed by using weighted nearest neighbors estimation. A modified estimator is proposed in Berrett et al. (2019):

$$\hat{H}_N^w = \frac{1}{N} \sum_{i=1}^{N} \sum_{j=1}^{k} w_i \log \xi_{(j),i}$$

where $w$ is the weight vector, $\xi_{(j),i} = e^{-\Psi(j)} \cdot c_1(n) \cdot (N-1) \cdot \rho_{(j),i}^d$, $\Psi$ denotes the digamma function. We choose the weight vector $w = (w_1, \ldots, w_k)$ as follows. For $k \in \mathbb{N}$ let

$$\mathcal{W}^{(k)} = \left\{ w \in \mathbb{R}^k : \sum_{j=1}^{k} w_j \cdot \frac{\Gamma\left(j + \frac{2\ell}{n}\right)}{\Gamma(j)} = 0 \text{ for } \ell = 1, \ldots, \left\lfloor \frac{n}{4} \right\rfloor, \right.$$

$$\left. \sum_{j=1}^{k} w_j = 1 \text{ and } w_j = 0 \text{ if } j \notin \left\{ \left\lfloor \frac{k}{n} \right\rfloor, \left\lfloor \frac{2k}{n} \right\rfloor, \ldots, k \right\} \right\}$$

and let the $w$ be a vector from $\mathcal{W}^{(k)}$ with the least $l_2$-norm.

Asymptotic: *the entropy estimation takes $\mathcal{O}(n \log n)$, weight selection – $\mathcal{O}(k^3) = \mathcal{O}(1)$, compression – $\mathcal{O}(n)$, resulting in a total time complexity of $\mathcal{O}(n \log n)$.*

### E.3 LIMITATIONS OF CLASSICAL ENTROPY ESTIMATORS

Although the entropy estimation is an example of a classical problem, it is still difficult to acquire estimates for high-dimensional data, as the estimation requires an exponentially (in dimension) large number of samples (see Goldfeld et al. (2020); McAllester & Stratos (2020)). As the mutual information estimation is tightly connected to the entropy estimation, this problem also manifests itself in our task. Although this difficulty affects every MI estimator, classical estimators may be assumed to be more prone to the curse of dimensionality, as they are usually too basic to grasp a manifold-like low-dimensional structure of high-dimensional data.

In this section, we provide experimental proofs of classical estimators' inability to yield correct MI estimates in the high-dimensional case. We utilize the same tests with images of 2D Gaussians used in Section 5 Figure 3. However, due to computational reasons, the size of the images is reduced to $4 \times 4$ and $8 \times 8$ (so the data is of even smaller dimension compared to Section 5). The results are presented int Table 2. For a comparison we also provide the results for WKL estimator fed with the PCA-compressed data.

Table 2: MSE (in nats) of classical MI estimation methods ran on $5 \cdot 10^3$ images of 2D Gaussians. The character "–" denotes cases, in which the method failed to work due computational reasons (numerical overflows, ill-conditioned matrices, etc.).

| Images size | KDE ML | KDE LSE | KL | WKL | WKL, PCA-compressed |
|---|---|---|---|---|---|
| $4 \times 4$ | $4.1 \cdot 10^3$ | $1,95 \cdot 10^1$ | $9,7$ | $2,9 \cdot 10^1$ | $1,87$ |
| $8 \times 8$ | $2.8 \cdot 10^3$ | – | $6,6 \cdot 10^1$ | $7,5 \cdot 10^1$ | $1,71$ |
| $16 \times 16$ | – | – | – | – | $0,67$ |

Note that although WKL estimator performs better in Section 5 due to lower bias, it is outperformed by the original KL estimator in the case of uncompressed data due to lower variance. However, this observation is not of great importance, as all the four methods perform poorly in the case of $8 \times 8$ images and bigger.

## F TECHNICAL DETAILS

In this section, we describe the technical details of our experimental setup: architecture of the neural networks, hyperparameters, etc.

For the tests described in Section 5, we use architectures listed in Table 3. The autoencoders are trained via Adam (Kingma & Ba, 2017) optimizer on $5 \cdot 10^3$ images with a batch size $5 \cdot 10^3$, a learning rate $10^{-3}$ and MAE loss for $2 \cdot 10^3$ epochs. The MINE critic network is trained via Adam optimizer on $5 \cdot 10^3$ images with a batch size $512$, a learning rate $10^{-3}$ for $5 \cdot 10^3$ epochs.

For the experiments described in Section 6, we use architectures listed in Table 4. The input data autoencoder is trained via Adam optimizer on $5 \cdot 10^4$ images with a batch size $1024$, a learning rate

Table 3: The NN architectures used to conduct the synthetic tests in Section 5.

| NN | Architecture |
|---|---|
| AEs, $16 \times 16$ ($32 \times 32$) images | $\times 1$: Conv2d(1, 4, ks=3), BatchNorm2d, LeakyReLU(0.2), MaxPool2d(2) |
| | $\times 1$: Conv2d(4, 8, ks=3), BatchNorm2d, LeakyReLU(0.2), MaxPool2d(2) |
| | $\times 2(3)$: Conv2d(8, 8, ks=3), BatchNorm2d, LeakyReLU(0.2), MaxPool2d(2) |
| | $\times 1$: Dense(8, dim), Tanh, Dense(dim, 8), LeakyReLU(0.2) |
| | $\times 2(3)$: Upsample(2), Conv2d(8, 8, ks=3), BatchNorm2d, LeakyReLU(0.2) |
| | $\times 1$: Upsample(2), Conv2d(8, 4, ks=3), BatchNorm2d, LeakyReLU(0.2) |
| | $\times 1$: Conv2d(4, 1, ks=3), BatchNorm2d, LeakyReLU(0.2) |
| MINE, critic NN, $16 \times 16$ ($32 \times 32$) images | $\times 1$: [Conv2d(1, 8, ks=3), MaxPool2d(2), LeakyReLU(0.01)]$^{\times 2 \text{ in parallel}}$ |
| | $\times 1(2)$: [Conv2d(8, 8, ks=3), MaxPool2d(2), LeakyReLU(0.01)]$^{\times 2 \text{ in parallel}}$ |
| | $\times 1$: Dense(128, 100), LeakyReLU(0.01) |
| | $\times 1$: Dense(100, 100), LeakyReLU(0.01) |
| | $\times 1$: Dense(100, 1) |

$10^{-3}$ and MAE loss for $2 \cdot 10^2$ epochs; the latent dimension equals $4$. The convolutional classifier is trained via Adam optimizer on $5 \cdot 10^4$ images with a batch size $1024$, a learning rate $10^{-5}$ and NLL loss for 300 epochs. The noise-to-signal ratio used for the Gaussian dropout is $10^{-3}$. Outputs of the layers are compressed via PCA into $4$-dimensional vectors. Mutual information is estimated via WKL estimator with $5$ nearest neighbours.

Table 4: The NN architectures used to conduct the information plane experiments in Section 6.

| NN | Architecture |
|---|---|
| Input data AE, $24 \times 24$ images | $\times 1$: Dropout(0.1), Conv2d(1, 8, ks=3), MaxPool2d(2), LeakyReLU(0.01) |
| | $\times 1$: Dropout(0.1), Conv2d(8, 16, ks=3), MaxPool2d(2), LeakyReLU(0.01) |
| | $\times 1$: Dropout(0.1), Conv2d(16, 32, ks=3), MaxPool2d(2), LeakyReLU(0.01) |
| | $\times 1$: Dense(288, 128), LeakyReLU(0.01) |
| | $\times 1$: Dense(128, dim), Sigmoid |
| CNN classifier | $L_1$: Conv2d(1, 8, ks=3), LeakyReLU(0.01) |
| | $L_2$: Conv2d(8, 16, ks=3), LeakyReLU(0.01) |
| | $L_3$: Conv2d(16, 32, ks=3), LeakyReLU(0.01) |
| | $L_4$: Dense(32, 32), LeakyReLU(0.01) |
| | $L_5$: Dense(32, 10), LogSoftMax |

Here we do not define $f_i$ and $g_i$ used in the tests with synthetic data, as these functions smoothly map low-dimensional vectors to high-dimensional images and, thus, are very complex. A Python implementation of the functions in question is available in the supplementary material, see the file `source/source/python/mutinfo/utils/synthetic.py`.

