# OpenReview forum: "Information Bottleneck Analysis of Deep Neural Networks via Lossy Compression"
_ICLR.cc/2024/Conference — ICLR 2024 poster_

### Official Review · Reviewer_Lyy3 · 2023-10-19

**Soundness:** 3 good
**Presentation:** 3 good
**Contribution:** 3 good
**Rating:** 6
**Confidence:** 1

**Summary:**

The Information Bottleneck (IB) principle provides an information-theoretic framework to analyze the training process of deep neural networks (DNNs) by tracking mutual information values. The training process involves two phases: fitting and compression, with compression believed to enhance generalization. The paper introduces an approach for IB analysis in general DNNs, overcoming challenges in estimating mutual information between high-dimensional vectors. The method involves compressing representations to estimate mutual information accurately. The approach is supported by theory and demonstrated through experiments, including analysis of a real-scale convolutional DNN, uncovering new insights into mutual information dynamics.

**Strengths:**

The theoretical analysis is sound and the experiments conducted in the paper are interesting and showing insides into the information flow of DNN.

**Weaknesses:**

Here are some comments / weaknesses / things to improve:

Abstract: „has only been verified for NNs of tiny sizes or specific types, such as quantized NNs“: I would not say that this is verified see summary table of [1]. The SNN paper did not saw a compression.

Introduction: „(a) stochastic NNs (Goldfeld et al., 2019; Tang Nguyen & Choi, 2019; Adilova et al., 2023) or (b) quantized NNs (Lorenzen et al., 2022).“: there is also the combination of both which was a paper at AAAI‘23 [2].

Statement 2: „This statement demonstrates that an arbitrary amount of information can be lost through compression. It arises from the fact that “less significant” in terms of metric spaces does not align with “less significant” in terms of information theory. However, with additional assumptions, a more useful theoretical result can be obtained.“: I am not sure but following the arguments from Goldfeld [3] MI is ill defined in deterministic DNN. So with this in mind no information can be removed in the compression phase. Can you maybe discuss this in the paper?


[1] Bernhard C. Geiger. On information plane analyses of neural network classifiers—a review. IEEE Transactions on Neural Networks and Learning Systems, 33(12):7039–7051, 12 2022. doi: 10.1109/ tnnls.2021.3089037.
[2] https://ojs.aaai.org/index.php/AAAI/article/download/25851/25623
[3] Ziv Goldfeld, Ewout van den Berg, Kristjan H. Greenewald, Igor V. Melnyk, Nam H. Nguyen, Brian Kingsbury, and Yury Polyanskiy. Estimating information flow in deep neural networks. In ICML, 2019.

**Questions:**

see weaknesses

---

> ### Author Response · Authors · 2023-11-17
>
> We sincerely thank Reviewer **Lyy3** for the work!
> We are more than glad to receive helpful critique of our article.
> We further provide answers to the problems and questions raised in the review.
> The references in our response correspond to the reference list in our manuscript.
>
> **Weaknesses:**
> 1. Indeed, Table 1 from Geiger (2020) shows that compression phase was observed in some (not all) experiments. For clarification, it is therefore better to replace “has only been verified” with “was only partially verified”. This will be done in the next revision.
> 2. We agree with the reviewer: the mentioned paper contains the analysis of quantized and stochastic NN's. We will add the citation in the next revision.
> 3. We kindly ask for clarification of the question, as in this particular part of our manuscript, we refer to the compression of data via an autoencoder. This should not be confused with the so-called “compression stage” in the “compression-fitting hypothesis”. We will change this part to “an arbitrary amount of information can be lost through compression **of the data**” in the next revision to avoid any potential confusion.
>
> We again thank Reviewer **Lyy3** for the constructive review! We hope that our answers helped to better understand the manuscript. Please feel free to let us know if there is anything specific we can provide or clarify. We are more than willing to answer any further questions and resolve any additional concerns. Looking forward to your response!

---

### Official Review · Reviewer_d5Zt · 2023-10-30

**Soundness:** 2 fair
**Presentation:** 3 good
**Contribution:** 2 fair
**Rating:** 5
**Confidence:** 3

**Summary:**

In this paper, authors have proposed a method for estimating the Shannon Mutual Information (MI) between the compressed representations of high-dimensional random vectors. Moreover, a discussion of information flow estimation based on the Information Bottleneck (IB) principle is presented for the MNIST classification problem using a simple LeNet architecture.

**Strengths:**

A systematic comparison between different entropy estimation methods in the MI estimation problem has been conducted in section 5.

**Weaknesses:**

While this paper tries to solve the well-known challenge in the IB principle, i.e., MI estimation for high dimensional vectors, I have some issues with the approach in this paper:

1 - I don't follow the claimed MI estimation approach. The proposed method has considered some of existing entropy estimation methods, (which they all have their own limitations) to estimate MI using the empirical frequencies in the MI expression (equation 1). What is the novelty of this approach ? The entropy estimation is a canonical problem in statistics. Both parametric and non-parametric have their own pros and cons. While non-parametric methods (e.g., KDE) do not consider any underlying assumption about the distribution of the data, they are typically not scalable in very high-dimensional regime (and they need a lot more training data), and they might be less sensitive than their parametric counterparts when the assumptions of the parametric methods are met.

2 - Another issues is the one that authors have also mentioned in the paper. Just providing a toy example of MNIST for a small classification model is not yet convincing to consider the scalability of the MI estimation approach, its accuracy on the high-dimensional data (not just 784), and its applicability in other non-classification problems (regression, unsupervised, generative, etc.).

3 - Some details about the functions $f_1$, $f_2$, $f$, $g$, etc do not exist in the paper.

**Questions:**

1 - Does the function $f$ in statement need to to be bijective (both injective and surjective) ? I think only being  injective is not sufficient.

---

> ### Author Response · Authors · 2023-11-16
>
> We thank Reviewer **d5Zt** for the work!
> We are glad to receive helpful critique of our article.
> We further provide answers to the main points raised in the review.
> The references in our response correspond to the reference list in our manuscript.
>
> **Weaknesses:**
> 1. Although the entropy estimation is an example of a classical problem, it is still problematic to acquire the estimates for high-dimensional data, as the estimation requires an exponentially (in dimension) large number of samples (see Goldfeld et al. (2020); McAllester \& Stratos (2020); $ \textnormal{estimation error} \in O(\textnormal{n.samples}^{-1/\textnormal{dim}}) $). The proposed approach makes the existing MI estimators less vulnerable to the curse of dimensionality by incorporating an explicit compression step. While other complex parametric MI estimators (like MINE (Belghazi et al., 2018)) may be assumed to compress data implicitly (as it is usually possible to extract some information about the PDF $ \rho_{X,Y} $ from the critic network), we argue that explicit compression allows for useful theoretical results and better estimation accuracy.
>
>    We also note that, unlike MI, entropy can be alternated via compression. That is why a theoretical analysis had to be conducted to justify the proposed approach.
>
> 2. Information bottleneck analysis of bigger or other non-classification problems (regression, generative or language modeling) is one of the possible directions for the future work on this topic. The current paper is devoted to the validation of the proposed method via a theoretical analysis, tests on synthetic data and application to some non-synthetic high-dimensional examples. Although the used problem is toy, it still poses difficulties in the task of MI estimation. That is why we believe our results to be a step forward.
>
> 3. The exact choice of $ f_i $ and $ g_i $ is not very important. In the main text, we only require $ f_i $ and $ g_i $ to satisfy the conditions of Statement 1 (see Algorithm 1; we will also add this requirement to Figure 2 in the next revision) to properly define the experimental setup. Defining $ f_i $ and $ g_i $ used in the tests requires a lot of space, as these functions smoothly map low-dimensional vectors to high-dimensional images and, thus, are very complex. A Python implementation of the functions in question is available in the supplementary material. We will consider adding the corresponding listings in the Appendix in the next revision.
>
> **Questions:**
> 1. If $ n' \neq n $, it is impossible to define $ f $ as a smooth bijective mapping between $ \mathbb{R}^{n'} $ and $ \mathbb{R}^n $. However, as $ f $ is injective, it is a bijection between $ \mathbb{R}^{n'} $ and $ f(\mathbb{R}^{n'}) \subseteq \mathbb{R}^n $. So $ f $ can be considered as a smooth bijection between $ \mathbb{R}^{n'} $ and a corresponding manifold in $ \mathbb{R}^n $.
>
> Finally, we thank Reviewer **d5Zt** for the valuable feedback. We are more than willing to address any further questions and will make every effort to resolve any additional concerns. Please feel free to let us know if there is anything specific we can provide or clarify. We hope that our answers helped to better understand the idea and contribution of our work. Looking forward to your response!

---

> ### Author Response · Authors · 2023-11-21
> **Awaiting your reply**
>
> Dear Reviewer **d5Zt**,
>
> Once again thank you very much for your detailed review and the time you spent. As we are nearing the end of the discussion period, we would like to ask if the questions you raised have been addressed. We hope you find our responses useful and would love to engage with you further if there are any remaining points.
>
> We understand that the discussion period is short, and we sincerely appreciate your time and help!

---

> > ### Comment · Reviewer_d5Zt · 2023-11-23
> >
> > Thanks for your answers. I am not still convinced about running a MNIST experiment. I am not saying to run on large models or generative models. I think other larger experiments need to be presented to demonstrate less vulnerability to the curse of dimensionality (as this is the main claim of the paper) and on different tasks (like regression) beyond simple digit classification. Many information bottleneck papers only include this dataset in the papers (although some have some experiments on CIFAR-10). So, this is my big concern even with the theoretical analysis provided in this paper.

---

### Official Review · Reviewer_MJhB · 2023-10-31

**Soundness:** 3 good
**Presentation:** 3 good
**Contribution:** 2 fair
**Rating:** 6
**Confidence:** 4

**Summary:**

The paper proposes to measure mutual information (MI) in neural networks via compressed representations, utilizing auto-encoder architectures that are assumed to be bijective mappings to a lower-dimensional space. With this rationale, the authors propose performing information plane analyses of stochastic neural networks using these compression-based estimators.

**Strengths:**

The paper is quite accessibly written and deals with an interesting topic. The results seem interesting, as they are not fully in line with what has commonly been observed in information plane analyses.

**Weaknesses:**

In addition to the questions below, I see a few weaknesses that prevent me from giving a better score:
- While it is clear that MI does not change under bijective mappings, the same may not hold for estimators of MI. In other words, I question the validity of the first equality in (4). To be more concrete, I think one can create counterexamples where a KDE of MI differs depending on whether it is obtained from the original data (dimension $n$) or the latent representation (dimension Sn'$). Since MI can only be estimated and not measured, this leads me to questioning the validity of the study: If compression affects the results of the estimator, then how can we be sure that the estimate represents the "true" (i.e., uncompressed) behavior of MI?
- Along that line, it would be important to see in Figs. 3 and 4 how the MI estimate obtained directly from the high-dimensional images behaves. This would more clearly show that compression is useful in comparison with estimators for high-dimensional data.

Summarizing, the paper does not show that the compressed representations are "better" for MI estimation than the original layer outputs/data, or that fundamentally different qualitative results can be obtained. While the approach seems reasonable and valid, I would like to see more evidence that it outperforms MI estimation from high-dimensional random variables.


Minor:
- In (5) and below, it should be made clear that $E_X$ is the encoder for $X$, while $E_Y$ is the encoder for $Y$.
- In the third paragraph of Section 5, the abbreviation WKL is not introduced.
- The colormap in Fig. 5 is not ideal, there is too little variation in the color.

**Questions:**

- In Fig. 3 and 4, why do the curves for most estimators decrease when compared to the true MI?
- In the same figures, why does MINE require a critic with the same architecture as the autoencoder, if its input are low-dimensional ($n'=2$ or $4$) signals? (Note that the MINE curves are blue.)
- In the same figures, there is good agreement between the blue and green curves. However, it can be assumed that the encoders $E_X$ and $E_Y$ act in a way to retrieve $f_1(\xi)$ and $g_1(\eta)$, respectively (or some other bijection thereof). With this in mind, it is not surprising that the curves agree so well. Has this been tested? For example, it could be a good idea to use a higher/smaller dimension for the output of the encoders as for the dimension of the generated signals.
- Why does the KL estimator have such a high offset?
- In Section 6, is noise also added during training or only during computing MI estimates? In any case, since noise is added one must not ignore the potential geometric affects that are induced by it (effects both during training and during estimation).
- How where the confidence intervals in Fig. 5 computed? How many networks have been trained?
- In Fig. 6, why is there so much variation/so large CIs at the end of training?
- In the same figure, why is there no decrease of "loss delta" at the end of training?
- In the same figure, does "loss delta" refer to the difference of losses between two consecutive epochs?
- In the same figure, the data processing inequality seems to be violated for $I(L;Y)$, see $L_2$ vs. $L_5$. Do you have an explanation for this?

_EDIT:_ After discussion with the authors, I have improved my score.

---

> ### Author Response · Authors · 2023-11-16
> **Official comment, Part I**
>
> Dear Reviewer **MJhB**, thank you sincerely for your careful and constructive review. We are glad to receive helpful critique.
> In the following text, we provide responses to your questions. We hope that all the concerns are properly addressed. The references in our response correspond to the reference list in our manuscript.
>
> **Weaknesses:**
> - It is, of course, true that a MI estimate obtained from the original data generally is not equal to an estimate obtained from the compressed representations; otherwise there is no point in compressing the data (except, probably, for computational complexity). However, conventional estimators completely fail to correctly estimate MI in high-dimensional cases due to the exponential sample complexity (see Goldfeld et al. (2020); McAllester \& Stratos (2020); $ \textnormal{estimation error} \in O(\textnormal{n.samples}^{-1/\textnormal{dim}}) $).
> It might be possible that equation (4) has been misunderstood: we do not claim that estimates for the original and compressed data are equal. Instead, we *define* MI estimate for high-dimensional compressible data through a conventional (e.g., via KDE) MI estimate for compressed representation.
> We will definitely improve this part of our work in the next revision to avoid such critical misunderstanding.
>
>    Our main theoretical and practical contribution is showing that the proposed MI estimator (defined by eq. (4)) is valid: we (a) show that MI is not lost via lossless compression, (b) bound MI loss in case of lossy compression, (c) provide experimental results with synthetic data showing good performance of the proposed method. One can be assured that the proposed estimation method represents the “true” behavior of MI via combining our results with a convergence analysis of the third-party estimator, which the compressed representations are fed into. In most of the cases, it can be done via the triangle inequality: $ |I(X;Y) - \hat I(E(X);Y)| \leq |I(X;Y) - I(E(X);Y)| + |I(E(X);Y) - \hat I(E(X);Y)| $, where the first term is analyzed in our article, and the second term should be acquired from a convergence analysis of a particular MI estimator.
>
> - It is a valid point that the contribution of our work would be more clear if we provide estimates computed directly from the data.
> However, as conventional entropy estimators exhibit bias and variance of such a high magnitude in case of high-dimensional data, it is impossible to plot results for the compressed and uncompressed data side by side without a complete loss of readability.
>
>    In particular, weighted Kozachenko-Leonenko (WKL) yields estimates ranging from about $ -3000 $ to $ 3000 $ nats with a variance around $ 2000 $ nats in the test with $ 8 \times 8 $ images of rectangles and the true value of MI varying from $ 0 $ to $ 10 $ nats. Other estimators yield the same nonsensical results for this $ 64 $-dimensional example. Thus, the error is of several orders of magnitude. In contrary, our method works fine even for the $ 32 \times 32 = 1024 $-dimensional data of the same kind.
>
>    One can reproduce these results via running the provided source code. We will add experimental proofs of the inability of the conventional methods to estimate MI of high-dimensional random vectors in the next revision.
>
> We hope that provided clarifications will help to better understand the manuscript and reevaluate the contribution of our work.
> We will add all the required evidence in the next revision to stress out that obtaining MI estimates via conventional estimators without compression is practically impossible.
> We will also correct all the mentioned minor weaknesses in the next revision.

---

> ### Author Response · Authors · 2023-11-16
> **Official comment, Part II**
>
> **Questions:**
> - In Figure 3 we attribute such behavior to the floating point error, as $ I(\xi;\eta) = 10 $ in case of $ n' = m' = 2 $ leads to correlation coefficient $ \sqrt{1 - e^{-10}} \approx 1 - 2 \cdot 10^{-5} $, yielding an ill-posed covariance matrix, which is then inverted on the first step of dataset generation.
>
>    In case of $ n' = m' = 4 $ (Figure 4) the correlation coefficient is $ \sqrt{1 - e^{-10/2}} \approx 1 - 3 \cdot 10^{-3} $,
>     and WKL yields better results. However, due to a doubled dimension, KDE and KL estimators exhibit an increased bias, as these methods are more sensitive to high dimensionality.
>
> - MINE is fed with uncompressed data. The choice of colors for this particular case is indeed misleading. Please refer to the text (“We also conduct experiments with MINE (without compression)”, Section 5, paragraph 3). We will change the color in the next revision. As MINE is fed with uncompressed data, the critic network should be at least as complex as the autoencoder to allow for fair comparison.
>
> - This hypothesis has not been tested. However, such good agreement may indeed indicate (almost) bijectivity of $ E_X \circ f_2 $ and $ E_Y \circ g_2 $. We will consider investigating this hypothesis in our future work.
>
> - KL estimator is proved to be biased for dimensions $ \ge 4 $, see Berrett et al. (2019).
>
> - Noise is present both during the training and the MI estimation. We kindly ask for clarification regarding potential geometric affects. Is this question related to the manifold hypothesis?
>
> - Confidence intervals are computed via a standard deviation derived from the entropy estimator. Around $ 35 $ networks were trained in total. For the final plot, only one network has been used.
>
> - We kindly ask for clarification of the question: all the networks used to acquire Figure 6 are autoencoders, which were trained before the MI evaluation on synthetic examples. Perhaps, the question is about Figure 5. In such case, the variation actually stayed almost the same throughout the training; we believe an increased density of points near the end of training might give the impression of an increased variance.
>
> - This happens because of the NN's loss reaching a plateau at the end of the training.
>
> - Yes, we refer to it as “loss function delta per epoch” at the end of page 9. We will define it more clearly in the next revision to avoid confusion.
>
> - As we plot estimates of MI, violation of DPI ranging within CIs is expected. If this violation is systematic, it may be attributed to more refined latent representations at the final layers of the network, which probably allow for a better MI estimation.
>
> Finally, we thank Reviewer **MJhB** again for the detailed constructive review. We are more than willing to address any further questions and will make every effort to resolve any additional concerns. Please feel free to let us know if there is anything specific we can provide or clarify. Looking for your response!

---

> > ### Comment · Reviewer_MJhB · 2023-11-20
> > **Thanks for the clarifications**
> >
> > Dear authors, thanks a lot for the clarifications. I indeed misread Fig. 5, I appreciate the explanations.
> >
> > ## Main Claim
> > I still have reservations about the main claims of the paper. For example, in response to my criticism:
> >
> > "If compression affects the results of the estimator, then how can we be sure that the estimate represents the "true" (i.e., uncompressed) behavior of MI?"
> >
> > you replied with the statement
> >
> > "One can be assured that the proposed estimation method represents the “true” behavior of MI via combining our results with a convergence analysis of the third-party estimator, which the compressed representations are fed into. [...]"
> >
> > This is an adequate response. At the same time, when discussing whether the DPI should hold or not, you respond with
> >
> > "As we plot estimates of MI, violation of DPI ranging within CIs is expected."
> >
> > To what extent can I now expect that the estimated values represent the qualitative behavior of MI?
> >
> > (Note that for $L_2$, at the end of training the CIs end below 1.8, while for $L_3$ they do not fall below 1.8 at the end of training -- the CIs do not overlap, so it is not a purely estimation-theoretic issue.)
> >
> > Of course, your estimates or the information plane analyses you conduct based on these estimates may still be _useful_. It is just worth reconsidering whether some of the claims in the paper (those about the connection between true MI and the MI estimate obtained from a compressed representation) should be weakened.
> >
> > ## Geometric effects
> > Adding Gaussian noise to intermediate representations pushes representations belonging to different classes apart from each other to reduce the "overlap" of the Gaussian distributions in latent space. Effectively, this may lead to a clustering of latent representations stronger than without added noise. Also, certain MI estimators rely on geometric effects (which is, e.g., quite obvious for binning and kernel density estimation, but may also be true for MINE if the critic has too little capacity).

---

> ### Author Response · Authors · 2023-11-20
>
> Dear Reviewer MJhB, thank you a lot for the reply.
>
> ### Main Claim
>
> It is our fault that we plot the CI obtained from entropy estimators only.
> That is why they represent only the error of a third-party estimator ran on compressed data, but not the information lost due to the data compression.
> We should change our experimental protocol to account for that. We must also change our claim from “the proposed estimation method represents the “true” behavior of MI...” to “the proposed estimation method represents the “true” behavior of MI **up to the information lost due to data compression, which can be bounded via Statement 3 and Corollary 6**...”. This claim should better represent the fact that our method is not ideal, and the estimation error can still be large in case of hard-to-compress data. This seems to happen in the case of $ L_2 $, as outputs of $ L_2 $ have dimension $ 400 $ and are linearly compressed to dimension $ 4 $; the reconstruction error is also relatively high compared to the consequent layers. We will investigate it further in the next revision.
>
> However, we still would like to stress the following:
> 1. Previous theoretical results (see Goldfeld et al. (2020); McAllester & Stratos (2020)) and our experimental results (Section E.3 in the new revision) show that estimation without compression is a much harder task, which is expected to result in estimates of a much lower quality.
> 2. If the compression is performed well, the estimate is close to the real value. It is demonstrated both via the experiments with synthetic data and the acquired bounds on the information lost due to the compression. Thanks to Statements 7,8,9 and Corollary 4, the information-theoretic quality of the compression can (under certain assumptions) be measured via the reconstruction error. We additionally note that the proposed bounds are tight, as they are derived from the data processing inequality.
> 3. In the work of Pool et al. (2019) it has been shown that other complex parametric NN-based estimators (NJW, JS, InfoNCE, etc.) exhibit poor performance during the estimation of MI between a pair of $ 20 $-dimensional _incompressible_ (i.e., not lying along a manifold) synthetic vectors. These vectors, however, are of much simpler structure than the synthetic datasets used in our work (Gaussian vectors and $ x \mapsto x^3 $ mapping applied to Gaussian vectors in (Pool et al., 2019) versus high-dimensional images of geometric shapes and functions in our work). We interpret these results as follows: if it is impossible to compress high-dimensional data to a lower dimension, not only our method fails, but the other modern parametric NN-based methods fail too.
>
> We hope that this information will (a) help to better understand the benefits of the explicit compression, (b) show that other modern and acknowledged methods exhibit the same limitations in case of incompressible data as our method do. Overall, judging by the tests performed in our work (Sections 5, C, E.3) and by the comprehensive overviews presented in (Poole et al., 2019; McAllester & Stratos, 2020), we think that our method is not flawless, but performs better than the existing MI estimators.
>
> ### Geometric effects
>
> Yes, these geometric effects are also mentioned in the work of Goldfeld et al. (2019). We do not ignore them, as we remember that the compression phase in the “fitting-compression hypothesis” can be caused by such clustering of the latent representations. However, keeping it in mind, no additional modifications to the overall experimental setup are required, as autoencoders should account for any peculiarities of the data while learning to reconstruct it. Moreover, in any case, (a) noise is widely used in modern networks for regularization purposes, (b) stochasticity still has to be introduced into non-quantized networks, otherwise it will be impossible to conduct information-theoretic analysis.
>
> We are sincerely thankful for the valuable feedback. We would be glad to continue the conversation if there are any questions left.

---

> > ### Comment · Reviewer_MJhB · 2023-11-21
> > **Thanks again**
> >
> > ad Main Claim: I agree with the modification of the claim, that makes sense. In general, I think it is important to stress more clearly that, as you say, "if it is impossible to compress high-dimensional data to a lower dimension, not only our method fails, but the other modern parametric NN-based methods fail too", and that some of the discrepancies are caused by exactly such phenomena (e.g., too strong compression for $L_2$.
> >
> > With these claims weakened, your approach is, in my opinion, a very useful one, and I will improve my score.

---

> ### Author Response · Authors · 2023-11-21
> **Thank you!**
>
> We are very grateful for the raised score! We again want to say thanks for the very careful and attentive reviewing of our paper, especially the critique of our graphics!

---

### Official Review · Reviewer_Fd1K · 2023-11-01

**Soundness:** 2 fair
**Presentation:** 3 good
**Contribution:** 2 fair
**Rating:** 6
**Confidence:** 3

**Summary:**

This paper proposed a framework for Information Bottleneck analysis of neural networks. To estimate the mutual information(MI) of high dimensional vectors, the paper proposes to use an auto-encoder to compress the data into a low dimensional space and provide a theoretical analysis of the error bounds. To avoid the hidden representation of a trained network $L$ becoming a deterministic function of $X$, the paper incorporates the Stochastic Neural Network. Numerical experiments on synthetic data show that the MI estimator of the compressed data can be close to the true value using the weighted Kozacheko-Leonenko estimator. Then the estimator is used on compressed data of several hidden layers $L_i$ of a convolutional network trained on MNIST to demonstrate the evolution of $I(L_i, X)$ and $I(L_i,Y)$.

**Strengths:**

Using information-theoretic analysis of DNNs to understand the training evolutions is an interesting topic. The proposed method provides a practical method to conduct information bottleneck analysis. Theoretical analysis and experiments on synthetic data demonstrate that an appropriate estimator on the compressed data can approximate the mutual information well.

**Weaknesses:**

I think a lot of details of the proposed method are missing.

1. What are the structures of the auto-encoder used to compress the data, and how are they trained? Is the result sensitive to the choice of the auto-encoder structure or some hyper-parameters? In algorithm 1, $E_X$, $E_Y$ of the auto-encoder is assumed to satisfy conditions of Corollary 1, so I think the auto-encoder is an important detail of the proposed method and should be discussed more in the paper or appendix. If the performance is sensitive to the details of auto-encoder choice and requires a lot of tuning, it will be a major drawback of the proposed method.

2. How are the stochastic NNs used in the proposed method, and what are the noises introduced in each layer? Incorporating stochastic NN is emphasized several times in the paper as a main component of the proposed method, but the details are missing.

3. Another weakness is that the proposed method is only applied to one real data example, the MNIST data. More experiments and analysis would improve the significance of the paper.

Readers are directed to the source code about these details, I think at least some more details of the proposed method should be provided in the paper or appendix.

**Questions:**

Please see the weakness part above, some minor questions

1. In section 3.1, it says "This hypothesis is believed to hold for a wide range of structured data, and there are datasets known to satisfy this assumption precisely." Any references to the hypothesis or some examples that the datasets satisfying this assumption precisely? I think some details would help understand the hypothesis.

2. The stochastic NN is introduced to help compute the mutual information, does It also affect model training? if not, what is the relationship of a stochastic NN and a regular trained NN?

---

> ### Author Response · Authors · 2023-11-14
> **Official comment, Part I**
>
> We thank Reviewer **Fd1K** for reading the article and providing us with a profound review!
> We further provide answers to the main points raised by the Reviewer.
> The references in our response correspond to the reference list in our manuscript.
>
> **Weaknesses:**
> - To compress synthetic data (Figures 3-4) and input data in experiments with DNN classifier, we use CNN autoencoders (AEs) trained to minimize MAE (yields less blurry results compared to MSE).
> We also use PCA — linear AE with MSE loss (allows for analytic expression of optimal compression mapping) to compress outputs of layers in DNN classifier experiment.
> We will add the requested details and more information on used networks to the Appendix in the next revision.
>
>    Statement 3 shows (under corresponding assumptions) the connection between loss of MI under compression and AE manifold learning capabilities (this connection is further investigated in Corollary 7 via deriving explicit bounds on loss of MI for PCA). Thus, the performance of the method is as sensitive to the AEs architecture as the reconstruction error is. Because reconstruction error is easy to track, AE architecture can be relatively easily validated or adjusted to match the needs of the task. We believe this makes our method more robust compared to other NN-based MI estimation approaches, where compression is assumed to be performed implicitly and thus cannot be validated (like MINE).
>
>    We also provide several results regarding much simpler architecture of AE — linear (PCA), see Appendix, Figure 6. As this figure shows, even PCA is capable of achieving almost the same quality as nonlinear AEs in the tests used in Section 5. One can consider it to be a somewhat practical evidence of low sensitivity to AE architecture. However, it is also possible to construct an adversarial example when the AE with insufficiently complex architecture fails (see Figure 6 (c)); such cases correspond to large reconstruction errors, which is a reliable signal to consider other AE architectures.
>
> - We explicitly mention stochasticity in Section 6: "To avoid the problem of a deterministic relationship between input and output, a small additive Gaussian noise is injected at each layer (we use a constant signal-to-noise ratio)". This setup is similar to Adilova et al. (2023), except we do not pass gradients through the noise variance. These details will also be emphasized in the Appendix in the next revision.
>
> - This is a valid point, and we are planning to perform more experiments with other neural networks and real datasets. However, we consider the proposed compression step and corresponding theoretical (bounds on MI under compression) and practical (tests on synthetic datasets with known MI) results to be the main contribution of our work. We use the experiment with the DNN classifier as a sanity check, trying to reproduce the compression-fitting pattern on the information plane.
>
> We again thank the Reviewer for pointing out weaknesses of our work. We will properly address them in the next revision.

---

> ### Author Response · Authors · 2023-11-14
> **Official comment, Part II**
>
> **Questions:**
> - In the introduction (page 2) we mention the work of Fefferman et al. (2013) [1]. This work provides a more formal and profound analysis of the manifold hypothesis (MH). We will add the citation to Section 3.1 as well.
>
>    One could think of photogrammetry (3D object reconstruction based on series of photos) as a good example of precise (up to imperfections in the setup) satisfaction of MH, as all the photos can be parametrized by a relative position and orientation of the camera. In particular, one can consider ENRICH dataset [2]. We also note NNs trained on such datasets, see [3].
>
> - Yes, stochasticity affects training of NN. Injecting random noise is widely considered as a way to increase model performance and generalization capabilities. For example, dropout layer [4] is widely used to regularize NNs. We also mention Gaussian dropout, which is known to benefit generalization even more than binary dropout [5]. Previous works on information bottleneck analysis also argue that using stochastic NNs does not have negative impact on training (see Goldfeld et al. (2019), Adilova et al. (2023)). In previous and in our work, stochastic NNs are used as a proxy to analyze deterministic NNs.
>
> The reasoning above will be incorporated into the article in the next revision to make it more accessible.
> We hope that we were able to provide sufficient answers to the mentioned concerns. We again thank Reviewer **Fd1K** for the helpful critique. We will be more than glad to continue the conversation if there are any questions left.
>
> **Additional references list:**
>
> [1] Fefferman et al. Testing the manifold hypothesis. Journal of the American Mathematical Society, 29, 10 2013. doi: 10.1090/jams/852.
> [2] Marelli et al. ENRICH: Multi-purposE dataset for beNchmaRking In Computer vision and pHotogrammetry. ISPRS Journal of Photogrammetry and Remote Sensing, v. 198, pp 84-98, 2023, doi: 10.1016/j.isprsjprs.2023.03.002
> [3] Mildenhall et al. NeRF: Representing Scenes as Neural Radiance Fields for View Synthesis. In: Vedaldi, A., Bischof, H., Brox, T., Frahm, JM. (eds) Computer Vision – ECCV 2020. ECCV 2020. Lecture Notes in Computer Science, vol 12346. Springer, Cham. doi: 10.1007/978-3-030-58452-8\_24
> [4] Geoffrey et al. Improving neural networks by preventing co-adaptation of feature detectors. arXiv:1207.0580
> [5] Srivastava et al. Dropout: a simple way to prevent neural networks from overfitting. Journal of Machine Learning Research, 15(1):1929–1958, 2014.

---

> > ### Comment · Reviewer_Fd1K · 2023-11-21
> >
> > Thank you for the detailed response and for adding details of the proposed method in the revision. I think the updated version makes the proposed method more accessible. I increased my score.

---

> ### Author Response · Authors · 2023-11-21
> **Awaiting your reply**
>
> Dear Reviewer **Fd1K**,
>
> Once again thank you very much for your detailed review and the time you spent. As we are nearing the end of the discussion period, we would like to ask if the questions you raised have been addressed. We hope you find our responses useful and would love to engage with you further if there are any remaining points.
>
> We understand that the discussion period is short, and we sincerely appreciate your time and help!

---

### Author Response · Authors · 2023-11-18
**Official comment regarding Rebuttal Revision**

Dear Reviewers,

We again sincerely thank you for reading the article and pointing out weak parts of our work. In the official comments below, we tried to answer all the concerns and questions. The new revision of the manuscript, which has been recently uploaded, is dedicated to improving the weak points mentioned in the reviews. We kindly invite the reviewers to read the corrected version of our article. We hope that we were able to properly address the majority of the concerns and problems that have been pointed out in the reviews.

Here we provide a brief list of the changes. The references correspond to the reference list in our manuscript.

1. We have added two new sections to the Appendix. Section E.3 contains proofs that classical entropy estimators fail in the high-dimensional case; so data compression is indeed required to properly estimate MI. Section F contains technical details of our experimental setup. We also note that Section F contains a path to a file in the supplementary materials with a Python 3 implementation of $ f_i $ and $ g_i $ used to generate synthetic datasets.  We hope that these amendments will help to better assess the contribution and reproducibility of our results.

2. Additional information about the manifold hypothesis (MH) has been added to Section 3.1, including an example of an area in computer vision, where one can find datasets satisfying the MH precisely.

3. Additional information about the role of stochasticity during the training of a NN has been added to the Introduction (the last paragraph at the end of p.2). Additional details about the type of noise used in our work has been added to the second paragraph in Section 6.

4. Some changes were introduced to make it less obscure that we define MI estimate via equations (4) and (5).

5. The colors of the plots have been adjusted to avoid confusion (Figure 2, 3, 4) and increase contrast (Figure 5).

6. The abstract has been slightly modified to better represent the results provided in (Geiger, 2022).

7. The citation of (Cerrato et al., 2023) has been added to the Introduction to provide an example of both stochastic and quantized NN.

8. Other minor changes to avoid confusion (e.g., proper introduction of abbreviations and variables).

We again thank the Reviewers for the work! We are more than willing to answer any further questions and resolve any additional concerns. Looking forward to your responses!

---

### Comment · Reviewer_Lyy3 · 2023-11-19
**Review stop**

Hello everybody,

on Friday I had an emergency medical intervention on my eyes and I am now unable to read for the next few weeks. This message is also written with an assistance.

Overall, I cannot further contribute to the review process.

All the best

---

### Author Response · Authors · 2023-11-23
**The second rebuttal revision**

Dear Reviewers,

We would like to thank you again for the time spent on reading our article and for the profound and constructive reviews you have provided.
As a result of the additional discussion with Reviewer **MJhB**, we decided to introduce the following changes in the next revision, which should be available now:

1. An additional paragraph has been added to the end of Section 3.2 to better illustrate the relation between the true dynamics of MI and the estimates yielded by the proposed method.
2. It is now clarified in the caption of Figure 5 that the confidence intervals represent errors of the MI estimator ran on the compressed data.
3. An additional paragraph has been added to the second item of the list in Section D to stress out that poor estimates for hard-to-compress data might be a universal problem.

We again sincerely thank the Reviewers for the valuable feedback.

---

### Meta-Review · Area_Chair_yGfy · 2023-12-05

**Metareview:**

The authors introduce a framework for information bottleneck analyses of larger neural networks and data dimensionalities. This involves using stochastic neural networks and an auto-encoder-based compression method. The reviewers generally acknowledged the importance of the topic and the authors’ thorough analyses. However, the reviewers raised questions about the validity of the mutual information estimation approach and the scale of the datasets. As the authors acknowledge, scaling this method beyond the now toy-ish MNIST and CIFAR-10 sized datasets would be interesting and further demonstrate the benefits of the proposed method.

**Justification For Why Not Higher Score:**

While the paper presents interesting results, the impact of the paper is currently limited by the size of the datasets tackled in the empirical analyses, which are now somewhat toy-ish in comparison with the modern literature.

**Justification For Why Not Lower Score:**

The paper presents an interesting approach to the well-known information bottleneck perspective, bringing this theory closer to the empirical settings that practitioners are currently exploring.

---

### Decision · Program_Chairs · 2024-01-16

Accept (poster)